# Deep Ridgelet Transform and Unified Universality Theorem for Deep and Shallow Joint-Group-Equivariant Machines

**Sho Sonoda** [1 2]  **Yuka Hashimoto** [3 1]  **Isao Ishikawa** [4 1]  **Masahiro Ikeda** [5 1]

## Abstract

We present a constructive universal approximation theorem for learning machines equipped with joint-group-equivariant feature maps, called the joint-equivariant machines, based on the group representation theory. "Constructive" here indicates that the distribution of parameters is given in a closed-form expression known as the ridgelet transform. Joint-group-equivariance encompasses a broad class of feature maps that generalize classical group-equivariance. Particularly, fully-connected networks are *not* group-equivariant *but* are joint-group-equivariant. Our main theorem also unifies the universal approximation theorems for both shallow and deep networks. Until this study, the universality of deep networks has been shown in a different manner from the universality of shallow networks, but our results discuss them on common ground. Now we can understand the approximation schemes of various learning machines in a unified manner. As applications, we show the constructive universal approximation properties of four examples: depth-$n$ joint-equivariant machine, depth-$n$ fully-connected network, depth-$n$ group-convolutional network, and a new depth-2 network with quadratic forms whose universality has not been known.

## 1. Introduction

One of the technical barriers in deep learning theory is that the relationship between parameters and functions is a black box. For this reason, the majority of authors build their theories on extremely simplified mathematical models. Such theories can explain the complex phenomena in deep learning only at a highly abstract level.

[1]RIKEN AIP [2]CyberAgent, Inc. [3]NTT Corporation [4]Kyoto University [5]The University of Osaka. Correspondence to: Sho Sonoda <sho.sonoda@riken.jp>.

*Proceedings of the 42$^{nd}$ International Conference on Machine Learning*, Vancouver, Canada. PMLR 267, 2025. Copyright 2025 by the author(s).

The proof of a universality theorem contains hints for understanding the internal data processing mechanisms inside neural networks. For example, the first universality theorem*s* for depth-2 neural networks were shown in 1989 with *four* different proofs by Cybenko (1989), Hornik et al. (1989), Funahashi (1989), and Carroll & Dickinson (1989). Among them, Cybenko and Hornik et al. presented existential proofs by using Hahn-Banach and Stone-Weierstrass respectively, meaning that it is not clear how to assign the parameters. On the other hand, Funahashi and Carroll-and-Dickinson presented constructive proofs by reducing networks to the Fourier transform and Radon transform respectively, meaning that it is clear how to assign the parameters. The latter constructive methods were refined as the so-called integral representation by Barron (1993) and further culminated as the *ridgelet transform*, the main objective of this study, discovered by Murata (1996) and Candès (1998).

To show the universality in a constructive manner, we formulate the problem as a functional equation: Let $\mathtt{M}[\gamma]$ denote a certain learning machine (such as a deep network) with parameter $\gamma$, and let $\mathcal{F}$ denote a class of functions to be expressed by the learning machine. Given a function $f \in \mathcal{F}$, find an unknown parameter $\gamma$ so that the machine $\mathtt{M}[\gamma]$ represents function $f$, i.e.

$$\mathtt{M}[\gamma] = f,$$

which we call a *learning equation*. This equation is a stronger formulation of learning than an ordinary formulation such as minimizing empirical risk $\sum_{i=1}^{n} |\mathtt{M}[\gamma](x_i) - f(x_i)|^2$ with respect to $\gamma$, as the latter is a weak form (or a variational form) of this equation. Therefore, characterizing the solution space of this equation leads to understanding the parameters obtained by risk minimization. Following previous studies (Murata, 1996; Candès, 1998; Sonoda et al., 2021a;b; 2022a;b), we call a solution operator $\mathtt{R}$ satisfying $\mathtt{M}[\mathtt{R}[f]] = f$ a *ridgelet transform*. Once such an $\mathtt{R}$ is found in a closed-form manner, we can present a constructive proof of *universality* because the reconstruction formula $\mathtt{M}[\mathtt{R}[f]] = f$ implies for any $f \in \mathcal{F}$ there exists a machine that implements $f$.

For depth-2 neural networks, the equation has been solved with several closed-form ridgelet transforms by using either Fourier expression method (Sonoda et al., 2024b), or group

representation method (Sonoda et al., 2024a). For example, the closed-form ridgelet transforms have been obtained for depth-2 fully-connected networks (Sonoda et al., 2021b), depth-2 fully-connected networks on manifolds (Sonoda et al., 2022b), depth-2 group convolution networks (Sonoda et al., 2022a), and depth-2 fully-connected networks on finite fields (Yamasaki et al., 2023). Furthermore, Sonoda et al. (2021a) have revealed that the distribution of parameters inside depth-2 fully-connected networks obtained by empirical risk minimization asymptotically converges to the ridgelet transform. In other words, the ridgelet transform can also explain the solutions obtained by risk minimization.

On the other hand, for depth-$n$ neural networks, the equation is far from solved, and it is common to either consider infinitely-deep mathematical models such as Neural ODEs (Sonoda & Murata, 2017b; E, 2017; Li & Hao, 2018; Haber & Ruthotto, 2017; Chen et al., 2018), or handcraft networks that approximate another universal approximators such as piecewise polynomial functions and indicator functions. For example, construction methods such as the Telgarsky sawtooth function (tent map, or the Yarotsky scheme) and bit extraction techniques (Cohen et al., 2016; Telgarsky, 2016; Yarotsky, 2017; 2018; Yarotsky & Zhevnerchuk, 2020; Daubechies et al., 2022; Cohen et al., 2022; Siegel, 2023; Petrova & Wojtaszczyk, 2023; Grohs et al., 2023) have been developed (not only to investigate the expressivity but also) to demonstrate the depth separation, super-convergence, and minmax optimality of deep ReLU networks. Various feature maps have also been handcrafted in the contexts of geometric deep learning (Bronstein et al., 2021) and deep narrow networks (Lu et al., 2017; Hanin & Sellke, 2017; Lin & Jegelka, 2018; Kidger & Lyons, 2020; Park et al., 2021; Li et al., 2023; Cai, 2023; Kim et al., 2024). However, for the purpose of understanding the parameters obtained by risk minimization, these results are less satisfactory because there is no guarantee that these handcrafted solutions are obtained by risk minimization in a manner presented by Sonoda et al. (2021a).

In order to investigate the relation between parameters and functions, we need to write down a general solution (i.e., the ridgelet transform) rather than handcrafting a particular solution. However, conventional ridgelet transforms have been limited to *depth-2* networks. In other words, existing methods cannot construct solutions for networks that repeatedly compose nonlinear activation functions more than twice—such as $\sigma(A_2\sigma(A_1\boldsymbol{x} - \boldsymbol{b}_1) - \boldsymbol{b}_2)$. In this study, inspired by the group-theoretic approach of Sonoda et al. (2024a), we derive the ridgelet transform for *depth-n* learning machines.

The contributions of this study are summarized as follows.

- We derive the ridgelet transform (solution operator for learning equation) for a general class of learning machines called the *joint-group-equivariant machine*

(Theorem 3.10), which shows the universal approximation theorem for a wide range of learning machines in a constructive and unified manner.

- As applications, we show the universal approximation properties of *four* examples: depth-$n$ joint-equivariant machine (Section 4), depth-$n$ fully-connected network (in Section 5), depth-$n$ group-convolutional network (in Section 6), and a new depth-2 network with quadratic forms whose universality has not been known (in Section 7).

Until this study, the universality of deep networks has been shown in a different manner from the universality of shallow networks, but our results discuss them on common ground. Now we can understand the approximation schemes of various learning machines in a unified manner.

## 2. Preliminaries

We quickly overview the original integral representation and the ridgelet transform, a mathematical model of depth-2 fully-connected network and its right inverse. Then, we list a few facts in the group representation theory. In particular, *Schur's lemma* play key roles in the proof of the main results.

**Notation.** For any topological space $X$, $C_c(X)$ denotes the Banach space of all compactly supported continuous functions on $X$. For any measure space $X$, $L^p(X)$ denotes the Banach space of all $p$-integrable functions on $X$. $\mathcal{S}(\mathbb{R}^d)$ and $\mathcal{S}'(\mathbb{R}^d)$ denote the classes of rapidly decreasing functions (or Schwartz test functions) and tempered distributions on $\mathbb{R}^d$, respectively.

### 2.1. Integral Representation and Ridgelet Transform for Depth-2 Fully-Connected Network

**Definition 2.1.** For any measurable functions $\sigma : \mathbb{R} \to \mathbb{C}$ and $\gamma : \mathbb{R}^m \times \mathbb{R} \to \mathbb{C}$, put

$$\mathtt{M}_\sigma[\gamma](\boldsymbol{x}) := \int_{\mathbb{R}^m \times \mathbb{R}} \gamma(\boldsymbol{a}, b)\sigma(\boldsymbol{a} \cdot \boldsymbol{x} - b)\mathrm{d}\boldsymbol{a}\mathrm{d}b, \ \boldsymbol{x} \in \mathbb{R}^m.$$

We call $\mathtt{M}_\sigma[\gamma]$ an (integral representation of) neural network, and $\gamma$ a parameter distribution.

The integration over all the hidden parameters $(\boldsymbol{a}, b) \in \mathbb{R}^m \times \mathbb{R}$ means all the neurons $\{\boldsymbol{x} \mapsto \sigma(\boldsymbol{a} \cdot \boldsymbol{x} - b) \mid (\boldsymbol{a}, b) \in \mathbb{R}^m \times \mathbb{R}\}$ are summed (or integrated, to be precise) with weight $\gamma$, hence formally $\mathtt{M}_\sigma[\gamma]$ is understood as a continuous neural network with a single hidden layer. We note, however, when $\gamma$ is a finite sum of point measures such as $\gamma_p = \sum_{i=1}^p c_i \delta_{(\boldsymbol{a}_i, b_i)}$ (by appropriately extending the class of $\gamma$ to Borel measures), then it can also reproduce a finite

width network

$$\mathtt{M}_\sigma[\gamma_p](\boldsymbol{x}) = \sum_{i=1}^p c_i \sigma(\boldsymbol{a}_i \cdot \boldsymbol{x} - b_i).$$

In other words, the integral representation is a mathematical model of depth-2 network with *any* width (ranging from finite to continuous).

Next, we introduce the ridgelet transform, which is known to be a right-inverse operator to $\mathtt{M}_\sigma$.

**Definition 2.2.** For any measurable functions $\rho : \mathbb{R} \to \mathbb{C}$ and $f : \mathbb{R}^m \to \mathbb{C}$, put

$$\mathtt{R}_\rho[f](\boldsymbol{a}, b) := \int_{\mathbb{R}^m} f(\boldsymbol{x}) \overline{\rho(\boldsymbol{a} \cdot \boldsymbol{x} - b)} \mathrm{d}\boldsymbol{x}, \ (\boldsymbol{a}, b) \in \mathbb{R}^m \times \mathbb{R}.$$

We call $\mathtt{R}_\rho$ a ridgelet transform.

To be precise, it satisfies the following reconstruction formula.

**Theorem 2.3** (Reconstruction Formula)**.** *Suppose $\sigma$ and $\rho$ are a tempered distribution $(\mathcal{S}')$ and a rapid decreasing function $(\mathcal{S})$ respectively. There exists a bilinear form $((\sigma, \rho))$ such that*

$$\mathtt{M}_\sigma \circ \mathtt{R}_\rho[f] = ((\sigma, \rho))f,$$

*for any square integrable function $f \in L^2(\mathbb{R}^m)$. Further, the bilinear form is given by $((\sigma, \rho)) = \int_{\mathbb{R}} \sigma^\sharp(\omega)\overline{\rho^\sharp(\omega)}|\omega|^{-m}\mathrm{d}\omega$, where $\sharp$ denotes the 1-dimensional Fourier transform.*

See Sonoda et al. (2021b, Theorem 6) for the proof. In particular, according to Sonoda et al. (2021b, Lemma 9), for any activation function $\sigma$, there always exists $\rho$ satisfying $((\sigma, \rho)) = 1$. Here, $\sigma$ being a tempered distribution means that typical activation functions are covered such as ReLU, step function, $\tanh$, gaussian, etc... We can interpret the reconstruction formula as a universality theorem of continuous neural networks, since for any given data generating function $f$, a network with output weight $\gamma_f = \mathtt{R}_\rho[f]$ reproduces $f$ (up to factor $((\sigma, \rho))$), i.e. $S[\gamma_f] = f$. In other words, the ridgelet transform indicates how the network parameters should be organized so that the network represents an individual function $f$.

The original ridgelet transform was discovered by Murata (1996) and Candès (1998). It is recently extended to a few modern networks by the Fourier slice method (see e.g. Sonoda et al., 2024b). In this study, we present a systematic scheme to find the ridgelet transform for a variety of given network architecture based on the group theoretic arguments.

## 2.2. Irreducible Representation and Schur's Lemma

In the main theorem, we use *Schur's lemma*, a fundamental theorem from group representation theory. We refer to Folland (2015) for more details on group representation and harmonic analysis on groups.

In this study, we assume group $G$ to be *locally compact*. This is a sufficient condition for having invariant measures. It is not a strong assumption. For example, any finite group, discrete group, compact group, and finite-dimensional Lie group are locally compact, while an *infinite*-dimensional Lie group is *not* locally compact.

Let $\mathcal{H}$ be a nonzero Hilbert space, and $\mathcal{U}(\mathcal{H})$ be the group of unitary operators on $\mathcal{H}$. A *unitary representation* $\pi$ of $G$ on $\mathcal{H}$ is a group homomorphism that is continuous with respect to the strong operator topology—that is, a map $\pi : G \to \mathcal{U}(\mathcal{H})$ satisfying $\pi_{gh} = \pi_g \pi_h$ and $\pi_{g^{-1}} = \pi_g^{-1}$, and for any $\psi \in \mathcal{H}$, the map $G \ni g \mapsto \pi_g[\psi] \in \mathcal{H}$ is continuous.

Suppose $\mathcal{M}$ is a closed subspace of $\mathcal{H}$. $\mathcal{M}$ is called an *invariant* subspace when $\pi_g[\mathcal{M}] \subset \mathcal{M}$ for all $g \in G$. Particularly, $\pi$ is called *irreducible* when it has only trivial invariant subspaces, namely $\{0\}$ and the whole space $\mathcal{H}$. The following theorem is a basic and useful characterization of irreducible representations.

**Theorem 2.4** (Schur's lemma)**.** *A unitary representation $\pi : G \to \mathcal{U}(\mathcal{H})$ is irreducible iff any bounded operator $T$ on $\mathcal{H}$ that commutes with $\pi$ is always a constant multiple of the identity. In other words, if $\pi_g \circ T = T \circ \pi_g$ for all $g \in G$, then $T = c\,\mathrm{Id}_\mathcal{H}$ for some $c \in \mathbb{C}$.*

See Folland (2015, Theorem 3.5(a)) for the proof. As we will see in the proof of the main theorem, an irreducible representation (or more generally, a *simple object*) is a standard unit for expressive power. Namely, suppose $X$ is a simple object (such as a simple abelian group, and an irreducible representation), and $N$ is a non-trivial sub-object (such as a normal group, and a sub-representation), then we can conclude $N = X$, which means the universality of $N$ in $X$. Schur's lemma restates this in terms of morphism. That is, the commutative property $\pi \circ T = T \circ \pi$ implies $T$ is a homomorphism, and thus it has to be either zero or identity.

As a concrete example of an irreducible representation, we use the following regular representation of the affine group $\mathrm{Aff}(m)$ on $L^2(\mathbb{R}^m)$.

**Theorem 2.5.** *Let $G := \mathrm{Aff}(m) := GL(m) \ltimes \mathbb{R}^m$ be the affine group acting on $X = \mathbb{R}^m$ by $(L, \boldsymbol{t}) \cdot \boldsymbol{x} = L\boldsymbol{x} + \boldsymbol{t}$, and let $\mathcal{H} := L^2(\mathbb{R}^m)$ be the Hilbert space of square-integrable functions. Let $\pi : \mathrm{Aff}(m) \to \mathcal{U}(L^2(\mathbb{R}^m))$ be the regular representation of the affine group $\mathrm{Aff}(m)$ on $L^2(\mathbb{R}^m)$, namely $\pi_g[f](\boldsymbol{x}) := |\det L|^{-1/2} f(L^{-1}(\boldsymbol{x} - \boldsymbol{t}))$ for any $g = (L, \boldsymbol{t}) \in G$. Then $\pi$ is irreducible.*

See Appendix E for the proof.

# 3. Main Results

We introduce unitary representations $\pi$ and $\widehat{\pi}$, a *joint-equivariant feature map* $\phi : X \times \Xi \to Y$, a *joint-equivariant machine* $\mathtt{M}[\gamma; \phi] : X \to Y$, and present the ridgelet transform $\mathtt{R}[f; \psi] : \Xi \to \mathbb{C}$ for joint-equivariant machines, yielding the universality $\mathtt{M}[\mathtt{R}[f; \psi]; \phi] = c_{\phi,\psi} f$. We note that $\pi$ plays a key role in the main theorem, and the joint-equivariance is an essential property of depth-$n$ fully-connected network.

Let $G$ be a locally compact group equipped with a left invariant measure $\mathrm{d}g$. Let $X$ and $\Xi$ be $G$-spaces equipped with $G$-invariant measures $\mathrm{d}x$ and $\mathrm{d}\xi$, called the *data domain* and the *parameter domain*, respectively. Let $Y$ be a separable Hilbert space, called the *output domain*. Let $\mathcal{U}(Y)$ be the space of unitary operators on $Y$, and let $\upsilon : G \to \mathcal{U}(Y)$ be a unitary representation of $G$ on $Y$. We call a $Y$-valued map $\phi$ on the data-parameter domain $X \times \Xi$, i.e. $\phi : X \times \Xi \to Y$, a *feature map*.

Let $L^2(X; Y)$ denote the space of $Y$-valued square-integrable functions on $X$ equipped with the inner product $\langle \phi, \psi \rangle_{L^2(X;Y)} := \int_X \langle \phi(x), \psi(x) \rangle_Y \mathrm{d}x$; and let $L^2(\Xi)$ denote the space of $\mathbb{C}$-valued square-integrable functions on $\Xi$.

If there is no risk of confusion, we use the same symbol $\cdot$ for the $G$-actions on $X$, $Y$, and $\Xi$ (e.g., $g \cdot x$, $g \cdot y$, and $g \cdot \xi$). On the other hand, to avoid the confusion between $G$-actions on output domain $Y$ and $Y$-valued function $f : X \to Y$, both "$g \cdot f(x)$" and "$\upsilon_g[f(x)]$" *always* imply $G$-action on $Y$, and "$\pi_g[f](x)$" (introduced soon below) for $G$-actions on $f : X \to Y$.

Additionally, we introduce two unitary representations $\pi$ and $\widehat{\pi}$ of $G$ on function spaces $L^2(X; Y)$ and $L^2(\Xi)$ as follows.

**Definition 3.1.** For each $g \in G$, $f \in L^2(X; Y)$ and $\gamma \in L^2(\Xi)$,

$$\pi_g[f](x) := \upsilon_g[f(g^{-1} \cdot x)] = g \cdot f(g^{-1} \cdot x), x \in X,$$

$$\widehat{\pi}_g[\gamma](\xi) := \gamma(g^{-1} \cdot \xi), \quad \xi \in \Xi.$$

In the main theorem, the irreducibility of $\pi$ will be a sufficient condition for the universality. On the other hand, the irreducibility of $\widehat{\pi}$ is not necessary. We have shown that $\pi$ and $\widehat{\pi}$ are unitary representations in Lemmas A.1 and A.2.

## 3.1. Joint-Equivariant Feature Map

We introduce the joint-group-equivariant feature map, extending the classical notion of group-equivariant feature maps. One of the major motivation to introduce this is that the depth-$n$ fully-connected network, the main subject of this study, is not equivariant but joint-equivariant.

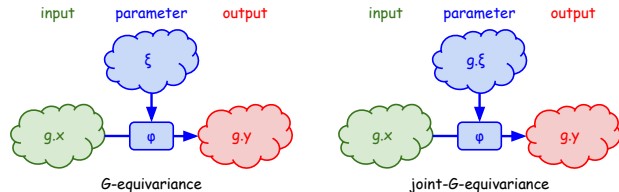

*Figure 1.* The classical $G$-equivariant feature map $\phi : X \times \Xi \to Y$ is a subclass of joint-$G$-equivariant map where the $G$-action on parameter domain $\Xi$ is *trivial*, i.e. $g \cdot \xi = \xi$

**Definition 3.2** (Joint-$G$-Equivariant Feature Map). We say a feature map $\phi : X \times \Xi \to Y$ is *joint-$G$-equivariant* when

$$\phi(g \cdot x, g \cdot \xi) = g \cdot \phi(x, \xi), \quad (x, \xi) \in X \times \Xi,$$

holds for all $g \in G$. Especially, when $G$-action on $Y$ is trivial, i.e. $\phi(g \cdot x, g \cdot \xi) = \phi(x, \xi)$, we say *joint-$G$-invariant*.

A basic example is the fully-connected layer

$$\phi(\boldsymbol{x}, (\boldsymbol{a}, b)) = \sigma(\boldsymbol{a} \cdot \boldsymbol{x} - b),$$

which is *not* $G$-equivariant *but* joint-$G$-equivariant. See Appendix B for more details.

*Remark* 3.3 (Relation to classical $G$-equivariance). The joint-$G$-equivariance is not a restriction but an extension of the classical notion of *$G$-equivariance*, i.e. $\phi(g \cdot x, \xi) = g \cdot \phi(x, \xi)$. In fact, $G$-equivariance is a special case of joint-$G$-equivariance where $G$ acts trivially on parameter domain, i.e. $g \cdot \xi = \xi$ (see Figure 1). Thus, all $G$-equivariant maps are automatically joint-$G$-equivariant.

### 3.1.1. Interpretation of Joint-Equivariant Maps

Obviously, $\phi$ is a $G$-map, namely a homomorphism between $G$-sets $X \times \Xi$ and $Y$. We denote the collection of all joint-$G$-equivariant maps as $\hom_G(X \times \Xi, Y)$. Equivalently, $\phi$ is identified with a $G$-map $\phi_c : \Xi \to Y^X$ through *currying* $\phi_c(\xi)(x) = \phi(x, \xi)$, satisfying $\phi_c(g \cdot \xi)(x) = \pi_g[\phi_c(\xi)](x)$. Further, $\phi$ is identified with the third $G$-map $\phi'_c : X \to Y^\Xi$ through $\phi'_c(x)(\xi) = \phi(x, \xi)$. These identifications are summarized as tensor-hom adjunction: $\hom_G(X \times \Xi, Y) \cong \hom_G(\Xi, Y^X) \cong \hom_G(X, Y^\Xi)$.

In terms of geometric deep learning, for example, Cohen et al. (2019) formulate the feature map as a vector field (or *section*). In their formulation, the joint-equivariant feature map $\phi_c : \Xi \to Y^X$ is understood as a global section of a trivial $G$-bundle $p : \Xi \times Y^X \to \Xi$ over base $\Xi$ with fiber $Y^X$, where structure group $G$ acts on fiber $Y^X$ by $\pi$.

We note, however, such geometric understanding is not unique. For example, in terms of learning equation $\mathtt{M} \circ \mathtt{R} = \mathrm{Id}_{Y^X}$, the learning machine $\mathtt{M} : \Xi \to Y^X$ is a feature map, and the ridgelet transform $\mathtt{R} : Y^X \to \Xi$ is a section (right-inverse). In this perspective, we can conversely understand the feature map $\phi_c : \Xi \to Y^X$ itself as a vector bundle (or *projection*) with base space $Y^X$ and total space $\Xi$.

3.1.2. CONSTRUCTION OF JOINT-EQUIVARIANT MAPS

In the following, we list several construction methods of joint-equivariant maps in Lemmas 3.4, 3.5 and 3.7 (in the next subsection), indicating the richness of the proposed concept. Whereas to construct a (non-joint) $G$-equivariant network, we must carefully and precisely design the network architecture (see, e.g., a textbook of geometric deep learning Bronstein et al., 2021), to construct a joint-$G$-equivariant network, we can easily and systematically obtain the one.

First, we can synthesize a joint-equivariant map from *any* map $\phi_0 : X \to Y$.

**Lemma 3.4.** *Let $X$ and $Y$ be $G$-sets. Fix an arbitrary map $\phi_0 : X \to Y$, and put $\phi(x, g) := \pi_g[\phi_0](x) = g \cdot \phi_0(g^{-1} \cdot x)$ for every $x \in X$ and $g \in G$. Then, $\phi : X \times G \to Y$ is joint-$G$-equivariant.*

*Proof.* For any $g, h \in G$, we have $\phi(g \cdot x, g \cdot h) = (gh) \cdot \phi_0((gh)^{-1} \cdot (g \cdot x)) = g \cdot \phi(x, h)$. □

In particular, the case of $X = Y = \Xi = G$, namely $\phi : G \times G \to G$, is understood as a primitive type of joint-$G$-equivariant maps.

The next lemma suggests the compatibility with function compositions, or deep structures.

**Lemma 3.5** (Depth-$n$ Joint-Equivariant Feature Map $\phi_{1:n}$). *Given a sequence of joint-$G$-equivariant feature maps $\phi_i : X_{i-1} \times \Xi_i \to X_i$ $(i = 1, \ldots, n)$, let $\Xi_{1:n} := \Xi_1 \times \cdots \times \Xi_n$ be the $n$-fold parameter space with the component-wise $G$-action $g \cdot \xi_{1:n} := (g \cdot \xi_1, \ldots, g \cdot \xi_n)$ for each $n$-fold parameters $\xi_{1:n} \in \Xi_{1:n}$, and let $\phi_{1:n} : X_0 \times \Xi_{1:n} \to X_n$ be the depth-$n$ feature map given by*

$$\phi_{1:n}(x, \xi_{1:n}) := \phi_n(\bullet, \xi_n) \circ \cdots \circ \phi_1(x, \xi_1).$$

*Then, $\phi_{1:n}$ is joint-$G$-equivariant.*

See Appendix A.2 for the proof. In other words, the composition of joint-equivariant maps defines a cascade product of morphisms: $\hom_G(\Xi_2, X_2^{X_1}) \times \hom_G(\Xi_1, X_1^{X_0}) \to \hom_G(\Xi_1 \times \Xi_2, X_2^{X_0})$.

### 3.2. Joint-Equivariant Machine

We introduce the joint-equivariant *machine*, extending the integral representation.

**Definition 3.6** (Joint-Equivariant Machine). Fix an arbitrary joint-equivariant feature map $\phi : X \times \Xi \to Y$. For any scalar-valued measurable function $\gamma : \Xi \to \mathbb{C}$, define a $Y$-valued map on $X$ by

$$\mathtt{M}[\gamma; \phi](x) := \int_\Xi \gamma(\xi)\phi(x, \xi)\mathrm{d}\xi, \quad x \in X,$$

where the integral is understood as the Bochner integral. We also write $\mathtt{M}_\phi := \mathtt{M}[\bullet; \phi]$ for short. If needed, we call the image $\mathtt{M}[\gamma; \phi] : X \to Y$ a joint-equivariant *machine*, and the integral transform $\mathtt{M}[\bullet; \phi]$ of $\gamma$ a joint-equivariant *transform*.

The joint-equivariant machine inherits the concept of the original integral representation—integrate all the available parameters $\xi$ to indirectly select which parameters to use by weighting on them, which *linearize* parametrization by lifting nonlinear parameters $\xi$ to linear parameter $\gamma$.

Moreover, the $G$-action $g \cdot \xi$ on parameter domain $\Xi$ is also linearized to linear representation $\widehat{\pi}$ of $G$ on $L^2(\Xi)$ (defined in Definition 3.1). As an important consequence, a *joint-$G$-equivariant machine $\mathtt{M}_\phi$ is joint-$G$-equivariant*. For later use, we formulate this slogan as the following formula.

**Lemma 3.7.** *Suppose $\phi : \Xi \to Y^X$ be joint-$G$-equivariant. Then, the associated joint-$G$-equivariant machine $\mathtt{M}_\phi : L^2(\Xi) \to L^2(X; Y)$ intertwines $\widehat{\pi}$ and $\pi$: For every $g \in G$, $\mathtt{M}_\phi \circ \widehat{\pi}_g = \pi_g \circ \mathtt{M}_\phi$.*

See Appendix A.3 for the proof. In other words, $\mathtt{M}$ is a functor from $\hom_G(\Xi, Y^X)$ to $\hom_G(L^2(\Xi), L^2(X; Y))$.

### 3.3. Ridgelet Transform

We introduce the ridgelet transform for joint-equivariant machines, extending the one for depth-2 fully-connected networks.

**Definition 3.8** (Ridgelet Transform). For any joint-equivariant feature map $\psi : X \times \Xi \to Y$ and $Y$-valued Borel measurable function $f$ on $X$, put a scalar-valued map by

$$\mathtt{R}[f; \psi](\xi) := \int_X \langle f(x), \psi(x, \xi) \rangle_Y \mathrm{d}x, \quad \xi \in \Xi.$$

We also write $\mathtt{R}_\psi := \mathtt{R}[\bullet; \psi]$ for short. If there is no risk of confusion, we call both the image $\mathtt{R}[f; \psi] : X \to Y$ and the integral transform $\mathtt{R}[\bullet; \psi]$ of $f$ a ridgelet transform.

Formally, it measures the similarity between target function $f$ and feature $\psi(\bullet, \xi)$ at $\xi$. As long as the integrals are convergent, the ridgelet transform is the dual operator of the joint-equivariant transform (with common $\phi$):

$$\langle \gamma, \mathtt{R}[f; \phi] \rangle_{L^2(\Xi)} = \int_{X \times \Xi} \gamma(\xi)\langle \phi(x, \xi), f(x) \rangle_Y \mathrm{d}x\mathrm{d}\xi$$

$$= \langle \mathtt{M}[\gamma; \phi], f \rangle_{L^2(X;Y)}.$$

As a dual statement for Lemma 3.7, the ridgelet transform is also joint-$G$-invariant and particularly an intertwiner.

**Lemma 3.9.** *Suppose $\psi \in \hom_G(\Xi, Y^X)$, then we have $\mathtt{R}_\psi \circ \pi_g = \widehat{\pi}_g \circ \mathtt{R}_\psi$ for every $g \in G$.*

In other words, $\mathtt{R}_\psi \in \hom_G(L^2(X; Y), L^2(\Xi))$. See Appendix A.4 for the proof.

## 3.4. Main Theorem

At last, we state the main theorem, that is, the reconstruction formula for joint-equivariant machines.

**Theorem 3.10** (Reconstruction Formula). *Assume (1) feature maps $\phi, \psi : X \times \Xi \to Y$ are joint-$G$-equivariant, (2) composite operator $\mathtt{M}_\phi \circ \mathtt{R}_\psi : L^2(X;Y) \to L^2(X;Y)$ is bounded (i.e., Lipschitz continuous), and (3) the unitary representation $\pi : G \to \mathcal{U}(L^2(X;Y))$ defined in Definition 3.1 is irreducible. Then, there exists a bilinear form $(\!(\phi, \psi)\!) \in \mathbb{C}$ (independent of $f$) such that for any $Y$-valued square-integrable function $f \in L^2(X;Y)$,*

$$\mathtt{M}_\phi[\mathtt{R}_\psi[f]] = \int_\Xi \int_X \langle f(x), \psi(x, \xi) \rangle_Y \mathrm{d}x \phi(\bullet, \xi) \mathrm{d}\xi = (\!(\phi, \psi)\!) f.$$

In practice, once the irreducibility of the representation $\pi$ on $L^2(X;Y)$ is verified, the ridgelet transform $\mathtt{R}_\psi$ becomes a right inverse operator of joint-equivariant transform $\mathtt{M}_\phi$ as long as $(\!(\phi, \psi)\!) \neq 0, \infty$. Despite the wide coverage of examples, the proof is brief and simple as follows.

*Proof.* Put $T := \mathtt{M}_\phi \circ \mathtt{R}_\psi : L^2(X;Y) \to L^2(X;Y)$. By Lemmas 3.7 and 3.9, $T$ commutes with $\pi$ as follows

$$\mathtt{M}_\phi \circ \mathtt{R}_\psi \circ \pi_g = \mathtt{M}_\phi \circ \widehat{\pi}_g \circ \mathtt{R}_\psi = \pi_g \circ \mathtt{M}_\phi \circ \mathtt{R}_\psi$$

for all $g \in G$. Hence by Schur's lemma (Theorem 2.4), there exist a constant $C_{\phi, \psi} \in \mathbb{C}$ such that $\mathtt{M}_\phi \circ \mathtt{R}_\psi = C_{\phi, \psi} \, \mathrm{Id}_{L^2(X)}$. Since $\mathtt{M}_\phi \circ \mathtt{R}_\psi$ is bilinear in $\phi$ and $\psi$, $C_{\phi, \psi}$ is bilinear in $\phi$ and $\psi$. □

*Remark* 3.11.   1. When $\pi$ is not irreducible (thus reducible) and admits an irreducible decomposition such as $L^2(X;Y) = \bigoplus_{i=1}^\infty \mathcal{H}_i$, then the reconstruction formula $\mathtt{M} \circ \mathtt{R}[f] = f$ holds for every $f \in \mathcal{H}_k$ for some $k$. This is another consequence from Schur's lemma.

2. The irreducibility is required only for $\pi$, and not for $\widehat{\pi}$. This asymmetry originates from the fact that our main theorem focuses on the universality of the learning machine, namely $\mathtt{M}_\phi[\gamma] : X \to Y$, not on its dual $\mathtt{R}_\psi[f] : \Xi \to \mathbb{R}$. When $\widehat{\pi}$ is irreducible, we can further conclude $\mathtt{R}_\psi \circ \mathtt{M}_\phi[\gamma] = \gamma$ for any $\gamma \in L^2(\Xi)$ (the order of composition is reverted from $\mathtt{M}_\phi \circ \mathtt{R}_\psi$). In practical examples such as fully-connected networks and wavelet analysis, however, $\mathtt{R}_\psi \circ \mathtt{M}_\phi$ is only a projection due to the redundancy of parameter distribution $\gamma(\boldsymbol{a}, b)$.

3. Relations to $cc$-universality, and sufficient conditions on $L^2$-boundedness of $\mathtt{M} \circ \mathtt{R}$, are discussed in Appendices C and D respectively.

4. As also mentioned in Section 3.1.1, $\mathtt{M}_\phi : L^2(\Xi) \to L^2(X;Y)$ is a $G$-equivariant vector bundle, and $\mathtt{R}_\psi : L^2(X;Y) \to L^2(\Xi)$ is a $G$-equivariant section.

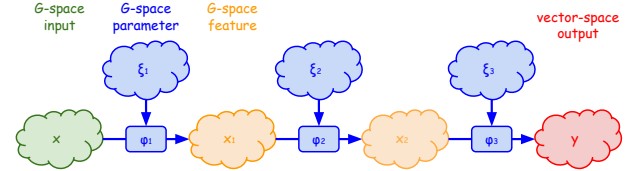

*Figure 2.* Deep $Y$-valued joint-$G$-equivariant machine on $G$-space $X$ is $L^2(X;Y)$-universal when unitary representation $\pi$ of $G$ on $L^2(X;Y)$ is irreducible, and the distribution of parameters for the machine to represent a given map $f : X \to Y$ is exactly given by the ridgelet transform $\mathtt{R}[f]$

5. The assumptions on feature maps $\phi, \psi$ that they are joint-equivariant and not orthogonal need to be verified in a case-by-case manner. Fortunately, we can use the closed-form expression of the ridgelet transform to our advantage. For example, for fully-connected networks (Section 5) and quadratic-form networks (Section 7), the joint-equivariance holds for any activation function. For the case of depth-2 fully-connected networks, it is known that the constant is zero if and only if the activation function is a polynomial function (see e.g., Sonoda & Murata, 2017a).

# 4. Example: Depth-$n$ Joint-Equivariant Machine

As pointed out in Lemma 3.5, the depth-$n$ feature map $\phi_{1:n}(x, \xi_{1:n}) = \phi_n(\bullet, \xi_n) \circ \cdots \circ \phi_1(x, \xi_1)$ is joint-$G$-equivariant when each component map $\phi_i$ is joint-equivariant. Hence, we can construct an $L^2(X;Y)$-universal deep joint-equivariant machine $\mathtt{DM}[\gamma; \phi_{1:n}]$ (see also Figure 2).

**Corollary 4.1** (Deep Ridgelet Transform). *For any maps $\gamma \in L^2(\Xi_{1:n})$ and $f \in L^2(X;Y)$, put*

$$\mathtt{DM}[\gamma; \phi_{1:n}](x) := \int_{\Xi_{1:n}} \gamma(\xi_{1:n}) \phi_{1:n}(x, \xi_{1:n}) \mathrm{d}\xi_{1:n}, \ x \in X,$$

$$\mathtt{R}[f; \psi_{1:n}](\xi_{1:n}) := \int_X \langle f(x), \psi_{1:n}(x, \xi_{1:n}) \rangle_Y \mathrm{d}x, \ \xi_{1:n} \in \Xi_{1:n}.$$

*Under the assumptions that $\mathtt{DM}_{\phi_{1:n}} \circ \mathtt{R}_{\psi_{1:n}}$ is bounded, and that $\pi$ is irreducible, there exists a bilinear form $(\!(\phi_{1:n}, \psi_{1:n})\!)$ satisfying $\mathtt{DM}_{\phi_{1:n}} \circ \mathtt{R}_{\psi_{1:n}} = (\!(\phi_{1:n}, \psi_{1:n})\!) \, \mathrm{Id}_{L^2(X;Y)}$.*

Again, it extends the original integral representation, and inherits the *linearization* trick of nonlinear parameters $\xi_{1:n}$ by integrating all the possible parameters (beyond the difference of layers) and indirectly select which parameters to use by weighting on them.

# 5. Example: Depth-$n$ Fully-Connected Network

We explain the case of depth-$n$ (precisely, depth-$n + 1$) fully-connected network.

Set $X = Y = \mathbb{R}^m$ (input and output domains), and for each $i \in \{1, \ldots, n\}$, set $X_i := \mathbb{R}^{d_i}$ (with $X_1 = X$ and $X_{n+1} = Y$), $\Xi_i := \mathbb{R}^{p_i \times d_i} \times \mathbb{R}^{p_i} \times \mathbb{S}_{d_{i+1}}^{q_i}$ (parameter domain), where $\mathbb{S}_d$ denotes the $d - 1$-dim. unit sphere, $\sigma_i : \mathbb{R}^{p_i} \to \mathbb{R}^{q_i}$ (activation functions), and define the feature map (vector-valued fully-connected neurons) as

$$\phi_i(\boldsymbol{x}_i, \boldsymbol{\xi}_i) := C_i \sigma_i(A_i \boldsymbol{x}_i - \boldsymbol{b}_i),$$

for every $\boldsymbol{x}_i \in \mathbb{R}^{d_i}, \boldsymbol{\xi}_i = (A_i, \boldsymbol{b}_i, C_i) \in \Xi_i$. Specifically, $d_1 = d_{n+1} = m$. If there is no risk of confusion, we omit writing $i$ for simplicity.

Let $O(m)$ denote the orthogonal group in dimension $m$. Let $G := O(m) \times \text{Aff}(m)$ be the product group of $O(m)$ and $\text{Aff}(m) = GL(m) \ltimes \mathbb{R}^m$. We suppose $G$ acts on the input and output domains as below: For any $g = (Q, L, \boldsymbol{t}) \in G = O(m) \times (GL(m) \ltimes \mathbb{R}^m)$,

$$g \cdot \boldsymbol{x} := L\boldsymbol{x} + \boldsymbol{t}, \ \boldsymbol{x} \in X, \quad g \cdot \boldsymbol{y} := v_g[\boldsymbol{y}] := Q\boldsymbol{y}, \ \boldsymbol{y} \in Y.$$

Namely, the group actions of both $O(m)$ on $X$ and $\text{Aff}(m)$ on $Y$ are trivial.

Let $\pi$ be the unitary representation of $G$ on the vector-valued square-integrable functions $\boldsymbol{f} \in L^2(X; Y)$, defined by

$$\pi_g[\boldsymbol{f}](\boldsymbol{x}) := |\det L|^{-1/2} Q\boldsymbol{f}(L^{-1}(\boldsymbol{x} - \boldsymbol{t})), \quad \boldsymbol{x} \in X$$

for each $g = (Q, L, \boldsymbol{t}) \in O(m) \times (GL(m) \ltimes \mathbb{R}^m)$.

**Lemma 5.1.** *The above $\pi : G \to \mathcal{U}(L^2(\mathbb{R}^m; \mathbb{R}^m))$ is irreducible.*

See Appendix A.5 for the proof. Additionally, we put the dual action of $G$ on parameter domain $\Xi_i$ as below:

$$g \cdot (A_i, \boldsymbol{b}_i, C_i) := \begin{cases} (A_i L^{-1}, \boldsymbol{b}_i + A_i L^{-1} \boldsymbol{t}, C_i), & i = 1 \\ (A_i, \boldsymbol{b}_i, C_i), & i \neq 1, n \\ (A_i, \boldsymbol{b}_i, Q C_i), & i = n \end{cases}$$

for all $g = (Q, L, \boldsymbol{t}) \in O(m) \times (GL(m) \ltimes \mathbb{R}^m)$, $(A_i, \boldsymbol{b}_i, C_i) \in \Xi_i$.

Then, the composition of feature maps $\phi_{1:n}(\boldsymbol{x}, \boldsymbol{\xi}_{1:n}) := \phi_n(\bullet, \boldsymbol{\xi}_n) \circ \cdots \circ \phi_1(\boldsymbol{x}, \boldsymbol{\xi}_1)$ is joint-$G$-equivariant. In fact,

$$\phi_1(g \cdot \boldsymbol{x}, g \cdot \boldsymbol{\xi}_1) = C_1 \sigma \left( A_1 L^{-1}(L\boldsymbol{x} + \boldsymbol{t}) - (\boldsymbol{b}_1 + A_1 L^{-1}\boldsymbol{t}) \right)$$
$$= C_1 \sigma(A_1 \boldsymbol{x} - \boldsymbol{b}_1) = \phi_1(\boldsymbol{x}, \boldsymbol{\xi}_1),$$
$$\phi_i(\boldsymbol{x}, g \cdot \boldsymbol{\xi}_i) = C_i \sigma(A_i \boldsymbol{x} - \boldsymbol{b}_i) = \phi_i(\boldsymbol{x}, \boldsymbol{\xi}_i), \quad i \neq 1, n$$
$$\phi_n(\boldsymbol{x}, g \cdot \boldsymbol{\xi}_n) = Q C_n \sigma(A_n \boldsymbol{x} - \boldsymbol{b}_n) = g \cdot \phi_n(\boldsymbol{x}, \boldsymbol{\xi}_n),$$

Therefore $\phi_{1:n}(g \cdot \boldsymbol{x}, g \cdot \boldsymbol{\xi}_{1:n}) = g \cdot \phi_{1:n}(\boldsymbol{x}, \boldsymbol{\xi}_{1:n})$.

So by putting depth-$n$ neural network and the corresponding ridgelet transform as below

$$\texttt{DNN}[\gamma; \phi_{1:n}](\boldsymbol{x}) = \int_{\Xi_{1:n}} \gamma(\boldsymbol{\xi}_{1:n}) \phi_{1:n}(\boldsymbol{x}, \boldsymbol{\xi}_{1:n}) \mathrm{d}\boldsymbol{\xi}_{1:n},$$

$$\texttt{R}[\boldsymbol{f}; \psi_{1:n}](\boldsymbol{\xi}_{1:n}) = \int_{\mathbb{R}^m} \boldsymbol{f}(\boldsymbol{x}) \cdot \overline{\psi_{1:n}(\boldsymbol{x}, \boldsymbol{\xi}_{1:n})} \mathrm{d}\boldsymbol{x},$$

Theorem 3.10 yields the reconstruction formula $\texttt{DNN}_{\phi_{1:n}} \circ \texttt{R}_{\psi_{1:n}} = ((\phi_{1:n}, \psi_{1:n})) \operatorname{Id}_{L^2(\mathbb{R}^m; \mathbb{R}^m)}$.

# 6. Example: Depth-$n$ Group Convolutional Network

As mentioned in Remark 3.3, all the classical equivariant feature maps, namely $\phi : X \times \Xi \to Y$ with trivial $G$-action on parameters: $\phi(g \cdot x, \xi) = g \cdot \phi(x, \xi)$, are automatically joint-equivariant. Therefore, once the irreducibility of representation $\pi$ is verified, our main theorem can state the ridgelet transform for classical $G$-equivariant networks.

In fact, in the case of group convolutional networks (GCNs) with *vector* inputs, we can reuse the irreducible representation for affine groups $\text{Aff}(m)$. In the following, we explain the ridgelet transform for *depth-$n$* GCNs, extending a general *depth-2* GCNs formulated by Sonoda et al. (2022a), which covers a wide range of typical group equivariant networks such as an ordinary $G$-convolution, DeepSets and E($n$)-equivariant maps in a unified manner.

In the previous study, the ridgelet transform was derived only for depth-2 GCNs, which is due to the proof technique based on the *Fourier expression method* (see Sonoda et al., 2024b, for more details), another proof technique for ridgelet transforms that does not require the irreducibility assumption but is limited to depth-2 learning machines.

In the following, we extend the GCNs from depth-2 to *depth-$n$* and derive the ridgelet transform by reviewing it from the group theoretic perspective. The main idea is to turn the depth-$n$ fully-connected network (FCN) $\phi_{1:n}$ in Section 5 to a depth-$n$ $G$-convolutional network, denoted $\phi_{1:n}^\tau$, by following the construction of the previous study.

## 6.1. Notations

Besides the primary group $G$ for convolution, we introduce an auxiliary group $A := O(m) \times \text{Aff}(m) = O(m) \times (GL(m) \ltimes \mathbb{R}^m)$, where $A$ and $G$ need not be homomorphic. Eventually, the irreducibility assumption of $\pi$ is required not for $G$ but for $A$. Hence, different from Section 5, the group acting on $L^2(X; Y)$ by $\pi$ is not $G$ but $A$. In accordance with the previous study, we write $T_g[\bullet]$ for $G$-action, $\alpha \cdot \bullet$ for $A$-action if needed, and $\tau_g[f](x) := T_g[f(T_{g^{-1}}[x])]$ for $G$-action on function $f : X \to Y$. By $L_G^2(X; Y)$, we denote the space of $G$-equivariant $Y$-valued functions

$f$ on $X$ that is square-integrable at the identity element $1_G$ of $G$, namely $L^2_G(X;Y) = \{f \in \hom_G(X,Y^G) \mid \|f(\bullet)(1_G)\|_{L^2(X;Y)} < \infty\} \cong \{\tau_\bullet[f_1] \mid f_1 \in L^2(X;Y)\}$.

From the next subsections, we will turn a joint-$A$-equivariant map $\phi_{1:n}$ to $G$-equivariant map $\phi^\tau_{1:n}$.

## 6.2. $G$-Convolutional Feature Map

For each $i$, let $\phi_i : X_i \times \Xi_i \to X_{i+1}$ be the fully-connected map $\phi_i(\boldsymbol{x}_i, \boldsymbol{\xi}_i) := C_i \sigma_i(A_i \boldsymbol{x}_i - \boldsymbol{b}_i)$ (as in Section 5). We define the $G$-convolutional map $\phi^\tau_i : X_i \times \Xi_i \to X^G_{i+1}$ as follows: For every $\boldsymbol{x}_i \in X_i$ and $\boldsymbol{\xi}_i = (A_i, \boldsymbol{b}_i, C_i) \in \Xi_i$,

$$\phi^\tau_i(\boldsymbol{x}_i, \boldsymbol{\xi}_i)(g) := \tau_g[\phi_i](\boldsymbol{x}_i, \boldsymbol{\xi}_i)$$
$$= T_g[(C_i \sigma_i(A_i T_{g^{-1}}[\boldsymbol{x}_i] - \boldsymbol{b}_i)], \quad g \in G.$$

By appropriately specifying the $G$-action $T$, the expression $A_i T_{g^{-1}}[\boldsymbol{x}_i]$ can reproduce a variety of general $G$-convolution products, say $a *_T x$, such as an ordinary $G$-convolution, the ones employed in DeepSets and E$(n)$-equivariant maps (see Section 5 of Sonoda et al., 2022a).

Similarly to Lemma 3.4, each $G$-convolutional map $\phi^\tau_i$ is $G$-equivariant in the classical sense because for any $g, h \in G$,

$$\phi^\tau_i(T_g[\boldsymbol{x}_i], \boldsymbol{\xi}_i)(h) = T_h[\phi_i(T_{h^{-1}}[T_g[\boldsymbol{x}_i]], \boldsymbol{\xi}_i)]$$
$$= T_g[T_{g^{-1}h}[\phi_i(T_{(g^{-1}h)^{-1}}[\boldsymbol{x}_i], \boldsymbol{\xi}_i)]] = \tau_g[\phi^\tau_i(\boldsymbol{x}_i, \boldsymbol{\xi}_i)](h).$$

Remarkably, the $G$-equivariance holds for any activation function $\sigma_i$, because it is applied element-wise in $G$.

## 6.3. $G$-Convolutional Network and Ridgelet Transform

Next, we define the depth-$n$ $G$-convolutional map $\phi^\tau_{1:n} : X \times \Xi_{1:n} \to Y^G$ by their compositions:

$$\phi^\tau_{1:n}(\boldsymbol{x}, \boldsymbol{\xi}_{1:n})(g) := \phi^\tau_n(\bullet, \boldsymbol{\xi}_n)(g) \circ \cdots \circ \phi^\tau_1(\boldsymbol{x}, \boldsymbol{\xi}_1)(g),$$

and define the depth-$n$ $G$-convolutional network and ridgelet transform as follows. For any $\gamma \in L^2(\Xi_{1:n})$ and $f \in L^2_G(X:Y)$,

$$\text{GCN}[\gamma; \phi^\tau_{1:n}](\boldsymbol{x})(g) := \int_{\Xi_{1:n}} \gamma(\boldsymbol{\xi}_{1:n}) \phi^\tau_{1:n}(\boldsymbol{x}, \boldsymbol{\xi}_{1:n})(g) \mathrm{d}\boldsymbol{\xi}_{1:n},$$

$$\text{R}_{\text{conv}}[f; \psi_{1:n}](\boldsymbol{\xi}_{1:n}) := \int_{\mathbb{R}^m} \langle f(\boldsymbol{x})(1_G), \psi_{1:n}(\boldsymbol{x}, \boldsymbol{\xi}_{1:n}) \rangle_Y \mathrm{d}\boldsymbol{x}.$$

See Appendix A.6 for more technical details on GCNs. The ridgelet transform encodes the information of function $f$ only at a single point $1_G$ (see also Lemma A.5). This is due to the $G$-equivariance of $f$ that the image at $g$ can be copied from the image at $1_G$ by translation: $f(\bullet)(g) = \tau_g[f|_{1_G}]$. In fact, the $G$-convolutions in depth-$n$ GCN has mechanism to expand the image at $1_G$ to entire $G$ by using $G$-equivariance (see Lemma A.4 for more precise meanings).

**Theorem 6.1** (Reconstruction Formula)*. There exists a bilinear form $((\phi_{1:n}, \psi_{1:n}))$ such that for any $f \in L^2_G(X:Y)$, $\text{GCN}[\text{R}_{\text{conv}}[f; \psi_{1:n}]; \phi^\tau_{1:n}] = ((\phi_{1:n}, \psi_{1:n}))f$.*

See Appendix A.7 for the proof. When $n = 2$, the argument here reproduces the one for depth-2 GCNs presented in Sonoda et al. (2022a). We remark that the base feature map $\phi$ and auxiliary group $A$ need not be the fully-connected network and affine group. In fact, we have never used the specific property of $C\sigma(A\boldsymbol{x} - \boldsymbol{b})$, but only used the group actions. Thus $A$ and $\phi$ can be arbitrary group and joint-$A$-equivariant map. When $A$ is the affine group, then the irreducibility of $\pi$ has already been verified in 5.1. On the other hand, when $A$ is another general group, we need to verify the irreducibility of representation $\pi$ of $A$ on $L^2(X;Y)$.

## 7. Example: Quadratic-form with Nonlinearity

Here, we present a new network for which the universality was not known.

Let $M$ denote the class of all $m \times m$-symmetric matrices equipped with the Lebesgue measure $\mathrm{d}A = \bigwedge_{i \geq j} \mathrm{d}a_{ij}$. Set $X = \mathbb{R}^m, \Xi = M \times \mathbb{R}^m \times \mathbb{R}$, and

$$\phi(\boldsymbol{x}, \xi) := \sigma(\boldsymbol{x}^\top A \boldsymbol{x} + \boldsymbol{x}^\top \boldsymbol{b} + c)$$

for any fixed function $\sigma : \mathbb{R} \to \mathbb{R}$. Namely, it is a quadratic-form in $x$ followed by nonlinear activation function $\sigma$.

Then, it is joint-invariant with $G = \text{Aff}(m)$ under the following group actions of $g = (\boldsymbol{t}, L) \in \mathbb{R}^m \rtimes GL(m)$:

$$(\boldsymbol{t}, L) \cdot \boldsymbol{x} := \boldsymbol{t} + L\boldsymbol{x},$$
$$(\boldsymbol{t}, L) \cdot (A, \boldsymbol{b}, c) := (L^{-\top} A L^{-1}, L^{-\top} \boldsymbol{b} - 2L^{-\top} A L^{-1} \boldsymbol{t},$$
$$c + \boldsymbol{t}^\top L^{-\top} A L^{-1} \boldsymbol{t} - \boldsymbol{t}^\top L^{-\top} \boldsymbol{b}).$$

See Appendix A.8 for the proof of joint-invariance. By Theorem 2.5, the regular representation $\pi$ of $\text{Aff}(m)$ on $L^2(\mathbb{R}^m)$ is irreducible. Hence as a consequence of the general result, the following network is $L^2(\mathbb{R}^m)$-universal.

$$\text{QNN}[\gamma](\boldsymbol{x}) := \int_\Xi \gamma(A, \boldsymbol{b}, c) \sigma(\boldsymbol{x}^\top A \boldsymbol{x} + \boldsymbol{x}^\top \boldsymbol{b} + c) \mathrm{d}A \mathrm{d}\boldsymbol{b} \mathrm{d}c.$$

## 8. Discussion

We have developed a systematic method for deriving a ridgelet transform for a wide range of learning machines defined by joint-group-equivariant feature maps, yielding the universal approximation theorems as corollaries. Traditionally, the techniques used in the expressive power analysis of deep networks were different from those used in the analysis of shallow networks, as overviewed in the introduction. Our main theorem unifies the approximation schemes of

both deep and shallow networks from the perspective of joint-group-action on the data-parameter domain. Technically, this unification is due to the irreducibility of group representations. From the traditional analytical viewpoint, universality refers to density. In this study, we have reviewed universality as irreducibility (or more generally, *simplicity* of objects) from an algebraic viewpoint. This switch of viewpoints has enabled us to reunify various universality theorems in a clear perspective.

**Additional Comments on Significance after Rebuttal**

The main theorem of this work is not merely a unification of existing results; a key strength lies in its ability to systematically and mechanically assess universality even for novel architectures. For example, it enables the straightforward design of expressive networks with universal approximation capability, such as quadratic networks.

In the era of large language models, the diversity of proposed architectures has significantly increased. Manually analyzing the expressive power of each new model on a case-by-case basis is no longer feasible from a theoretical standpoint. Our framework addresses this challenge by offering a general principle.

In particular, algebraic tools such as Schur's lemma provide a notable advantage in that they allow us to assess universality (or density) even when the inputs and outputs are not vectors—something that traditional techniques often struggle with. While classical theorems like the Stone–Weierstrass or Hahn–Banach theorems offer generic tools for establishing universality, applying them to deep neural networks with hierarchical structures requires considerable ingenuity.

By contrast, proof techniques originating with Hornik et al. (1989), which rely on repeatedly differentiating activation functions to construct polynomial approximations, often obscure the intuitive mechanism by which neural networks perform approximation. In comparison, our use of joint group equivariance and ridgelet transforms as inverse operators provides a more direct and interpretable understanding of the approximation process in deep networks.

Regarding generalization: Theoretical analysis of generalization error is often based on estimates of the Rademacher complexity of the hypothesis class. However, in deep learning, the Rademacher complexity typically scales exponentially with network depth. This is at odds with practical observations, where deeper networks tend to generalize better.

This discrepancy arises from the limitations of current techniques for analyzing composite functions: the reliance on coarse Lipschitz estimates leads to overly pessimistic bounds. To address this gap, we revisited expressive power analysis and developed the deep ridgelet transform as a theoretical framework that can precisely handle compositions of functions.

## Acknowledgements

This work was supported by JSPS KAKENHI 24K21316, 25H01453, JST PRESTO JPMJPR2125, JST BOOST JP-MJBY24E2, JST CREST JPMJCR2015 and JPMJCR1913.

## Impact Statement

This paper presents work whose goal is to advance the field of Machine Learning. There are many potential societal consequences of our work, none which we feel must be specifically highlighted here.

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

# A. Proofs

## A.1. Unitarity of Representations

In Definition 3.1, $\pi$ and $\widehat{\pi}$ are defined as below: For each $g \in G$, $f \in L^2(X;Y)$ and $\gamma \in L^2(\Xi)$,

$$\pi_g[f](x) := \upsilon_g[f(g^{-1} \cdot x)] = g \cdot f(g^{-1} \cdot x),$$
$$\widehat{\pi}_g[\gamma](\xi) := \gamma(g^{-1} \cdot \xi).$$

**Lemma A.1.** $\pi$ *is a unitary representation of $G$ on $L^2(X;Y)$.*

*Proof.* Recall that the representation $\upsilon$ of $G$ on $Y$ is unitary. So, for any $g, h \in G$ and $f \in L^2(X;Y)$,

$$\pi_g[\pi_h[f]](x) = g \cdot (h \cdot f(h^{-1} \cdot (g^{-1} \cdot x))) = (gh) \cdot f((gh)^{-1} \cdot x) = \pi_{gh}[f](x),$$

and for any $g \in G$ and $f_1, f_2 \in L^2(X;Y)$,

$$\langle \pi_g[f_1], \pi_g[f_2] \rangle_{L^2(X;Y)} = \int_X \langle \upsilon_g[f_1(g^{-1} \cdot x)], \upsilon_g[f_2(g^{-1} \cdot x)] \rangle_Y \mathrm{d}x$$
$$= \int_X \langle f_1(x), \upsilon_g^*[\upsilon_g[f_2(x)]] \rangle_Y \mathrm{d}x = \langle f_1, f_2 \rangle_{L^2(X;Y)}. \qquad \square$$

**Lemma A.2.** $\widehat{\pi}$ *is a unitary representation of $G$ on $L^2(\Xi)$.*

*Proof.* For any $g, h \in G$ and $\gamma \in L^2(\Xi)$,

$$\widehat{\pi}_g[\widehat{\pi}_h[\gamma]](\xi) = \gamma(h^{-1} \cdot (g^{-1} \cdot \xi) = \gamma((gh)^{-1} \cdot \xi) = \widehat{\pi}_{gh}[f](x),$$

and for any $g \in G$ and $\gamma_1, \gamma_2 \in L^2(\Xi)$,

$$\langle \widehat{\pi}_g[\gamma_1], \widehat{\pi}_g[\gamma_2] \rangle_{L^2(\Xi)} = \int_\Xi \gamma_1(g^{-1} \cdot \xi)\overline{\gamma_2(g^{-1} \cdot \xi)}\mathrm{d}\xi$$
$$= \int_\Xi \gamma_1(\xi)\overline{\gamma_2(\xi)}\mathrm{d}\xi = \langle \gamma_1, \gamma_2 \rangle_{L^2(\Xi)}. \qquad \square$$

## A.2. Proof of Lemma 3.5

*Proof.* For any $g \in G, x \in X$, and $\xi_{1:n} \in \Xi_{1:n}$, we have

$$\phi_{1:n}(g \cdot x, g \cdot \xi_{1:n}) = \phi_n(\bullet, g \cdot \xi_n) \circ \cdots \circ \phi_2(\bullet, g \cdot \xi_2) \circ \phi_1(g \cdot x, g \cdot \xi_1)$$
$$= \phi_n(\bullet, g \cdot \xi_n) \circ \cdots \circ \phi_2(g \cdot \bullet, g \cdot \xi_2) \circ \phi_1(x, \xi_1)$$
$$\vdots$$
$$= \phi_n(g \cdot \bullet, g \cdot \xi_n) \circ \cdots \circ \phi_2(\bullet, \xi_2) \circ \phi_1(x, \xi_1)$$
$$= g \cdot \phi_n(\bullet, \xi_n) \circ \cdots \circ \phi_2(\bullet, \xi_2) \circ \phi_1(x, \xi_1)$$
$$= g \cdot \phi_{1:n}(x, \xi_{1:n}). \qquad \square$$

## A.3. Proof of Lemma 3.7

*Proof.* We use the left-invariance of measure $\mathrm{d}\xi$, and joint-$G$-equivariance of $\phi : X \times \Xi \to Y$. For any $g \in G, x \in X$, we have

$$\mathtt{M}_\phi[\widehat{\pi}_g[\gamma]](x) = \int_\Xi \gamma(g^{-1} \cdot \xi)\phi(x, \xi)\mathrm{d}\xi$$
$$= \int_\Xi \gamma(\xi)\phi(x, g \cdot \xi)\mathrm{d}\xi$$
$$= \int_\Xi \gamma(\xi)\upsilon_g[\phi(g^{-1} \cdot x, \xi)]\mathrm{d}\xi = \pi_g[\mathtt{M}_\phi[\gamma]](x). \qquad \square$$

## A.4. Proof of Lemma 3.9

*Proof.* We use the unitarity of representation $\upsilon : G \to \mathcal{U}(Y)$, left-invariance of measure $\mathrm{d}x$, and joint-$G$-equivariance of $\psi : X \times \Xi \to Y$. For any $g \in G, \xi \in \Xi$, we have

$$
\begin{aligned}
\mathrm{R}_\psi[\pi_g[f]](\xi) &= \int_X \langle \upsilon_g[f(g^{-1} \cdot x)], \psi(x, \xi) \rangle_Y \mathrm{d}x \\
&= \int_X \langle f(g^{-1} \cdot x), \upsilon_g^*[\psi(x, \xi)] \rangle_Y \mathrm{d}x \\
&= \int_X \langle f(x), \upsilon_g^*[\psi(g \cdot x, \xi)] \rangle_Y \mathrm{d}x \\
&= \int_X \langle f(x), \psi(x, g^{-1} \cdot \xi) \rangle_Y \mathrm{d}x = \widehat{\pi}_g[\mathrm{R}_\psi[f]](\xi). \qquad \square
\end{aligned}
$$

## A.5. Proof of Lemma 5.1

*Proof.* We use the following fact.

**Lemma A.3** (Folland (2015, Theorem 7.12)). *Let $\pi_1$ and $\pi_2$ be representations of locally compact groups $G_1$ and $G_2$, and let $\pi_1 \otimes \pi_2$ be their outer tensor product, which is a representation of the product group $G_1 \times G_2$. Then, $\pi_1$ and $\pi_2$ are irreducible if and only if $\pi_1 \otimes \pi_2$ is irreducible.*

Recall the representations of $O(m)$ on $\mathbb{R}^m$ and of $\mathrm{Aff}(m)$ on $L^2(\mathbb{R}^m)$ are respectively irreducible (see Theorem 2.5), and $L^2(\mathbb{R}^m; \mathbb{R}^m)$ is equivalent to the tensor product $\mathbb{R}^m \otimes L^2(\mathbb{R}^m)$. Hence by Lemma A.3, the representation $\pi$ of the product group $O(m) \times \mathrm{Aff}(m)$ on the tensor product $\mathbb{R}^m \otimes L^2(\mathbb{R}^m) = L^2(\mathbb{R}^m; \mathbb{R}^m)$ is irreducible. $\qquad \square$

## A.6. Connection between GCN and FCN

Recall that the depth-$n$ FCN and its ridgelet transform introduced in Section 5 are given as below.

$$
\mathrm{DNN}[\gamma; \phi_{1:n}](\boldsymbol{x}) := \int_{\Xi_{1:n}} \gamma(\boldsymbol{\xi}_{1:n}) \phi_{1:n}(\boldsymbol{x}, \boldsymbol{\xi}_{1:n}) \mathrm{d}\boldsymbol{\xi}_{1:n},
$$

$$
\mathrm{R}_{\mathtt{fc}}[f; \psi_{1:n}](\boldsymbol{\xi}_{1:n}) = \int_{\mathbb{R}^m} \langle f(\boldsymbol{x}), \psi_{1:n}(\boldsymbol{x}, \boldsymbol{\xi}_{1:n}) \rangle_Y \mathrm{d}\boldsymbol{x}.
$$

As a consequence of Lemmas 3.5 and 3.7, we have the following.

**Lemma A.4.** $\mathrm{GCN}[\gamma; \phi_{1:n}^\tau](\boldsymbol{x})(g) = \tau_g[\mathrm{DNN}[\gamma; \phi_{1:n}]](\boldsymbol{x})$.

*Proof.*

$$
\begin{aligned}
\phi_{1:n}^\tau(\boldsymbol{x}, \boldsymbol{\xi}_{1:n})(g) &= T_g[\phi_n(\bullet, \boldsymbol{\xi}_n) \circ \cdots \circ \phi_1(T_{g^{-1}}[\boldsymbol{x}], \boldsymbol{\xi}_1)] \\
&= T_g[\phi_{1:n}(T_{g^{-1}}[\boldsymbol{x}], \boldsymbol{\xi}_{1:n})] \\
&= \tau_g[\phi_{1:n}](\boldsymbol{x}, \boldsymbol{\xi}_{1:n}),
\end{aligned}
$$

and thus

$$
\mathrm{GCN}[\gamma; \phi_{1:n}^\tau](\boldsymbol{x})(g) = \int_{\Xi_{1:n}} \gamma(\boldsymbol{\xi}_{1:n}) \tau_g[\phi_{1:n}](\boldsymbol{x}, \boldsymbol{\xi}_{1:n}) \mathrm{d}\boldsymbol{\xi}_{1:n} = \tau_g[\mathrm{DNN}[\gamma; \phi_{1:n}]](\boldsymbol{x}). \qquad \square
$$

**Lemma A.5.** $\mathrm{R}_{\mathtt{conv}}[f; \psi_{1:n}](\boldsymbol{\xi}_{1:n}) = \mathrm{R}_{\mathtt{fc}}[f(\bullet)(1_G); \psi_{1:n}](\boldsymbol{\xi}_{1:n})$.

*Proof.* Immediate from the definition. $\qquad \square$

## A.7. Proof of Theorem 6.1

*Proof.* By Lemmas A.4 and A.5,

$$
\begin{aligned}
\texttt{GCN}[\texttt{R}_{\texttt{conv}}[f;\psi_{1:n}];\phi_{1:n}^{\tau}](\boldsymbol{x})(g) &= \tau_g[\texttt{DNN}[\texttt{R}_{\texttt{fc}}[f(\bullet)(1_G);\psi_{1:n}];\phi_{1:n}]](\boldsymbol{x}) \\
&= \tau_g[(\!(\phi_{1:n},\psi_{1:n})\!)f(\bullet)(1_G)](\boldsymbol{x}) \\
&= (\!(\phi_{1:n},\psi_{1:n})\!)f(\boldsymbol{x})(g).
\end{aligned}
$$

$\square$

## A.8. Joint-equivariance of quadratic-form network

The feature map and group actions are given as follows.

$$
\begin{aligned}
\phi(\boldsymbol{x},\xi) &:= \sigma(\boldsymbol{x}^{\top}A\boldsymbol{x}+\boldsymbol{x}^{\top}\boldsymbol{b}+c), \\
(\boldsymbol{t},L)\cdot\boldsymbol{x} &:= \boldsymbol{t}+L\boldsymbol{x}, \\
(\boldsymbol{t},L)\cdot(A,\boldsymbol{b},c) &:= (L^{-\top}AL^{-1}, L^{-\top}\boldsymbol{b}-2L^{-\top}AL^{-1}\boldsymbol{t}, c+\boldsymbol{t}^{\top}L^{-\top}AL^{-1}\boldsymbol{t}-\boldsymbol{t}^{\top}L^{-\top}\boldsymbol{b}).
\end{aligned}
$$

Then, it is joint-invariant. In fact,

$$
\begin{aligned}
\phi(g\cdot\boldsymbol{x},g\cdot\boldsymbol{\xi}) &= \sigma((L\boldsymbol{x}+\boldsymbol{t})^{\top}L^{-\top}AL^{-1}(L\boldsymbol{x}+\boldsymbol{t})+(L\boldsymbol{x}+\boldsymbol{t})^{\top}(L^{-\top}\boldsymbol{b}-2L^{-\top}AL^{-1}\boldsymbol{t})+...) \\
&= \sigma(\boldsymbol{x}^{\top}A\boldsymbol{x}+2\boldsymbol{x}^{\top}AL^{-1}\boldsymbol{t}+\boldsymbol{t}^{\top}L^{-\top}AL^{-1}\boldsymbol{t}+\boldsymbol{x}^{\top}\boldsymbol{b}-2\boldsymbol{x}^{\top}AL^{-1}\boldsymbol{t}+\boldsymbol{t}^{\top}L^{-\top}\boldsymbol{b} \\
&\qquad -2\boldsymbol{t}^{\top}L^{-\top}AL^{-1}\boldsymbol{t}+c+\boldsymbol{t}^{\top}L^{-\top}AL^{-1}\boldsymbol{t}-\boldsymbol{t}^{\top}L^{-\top}\boldsymbol{b}) \\
&= \sigma(\boldsymbol{x}^{\top}A\boldsymbol{x}+\boldsymbol{x}^{\top}\boldsymbol{b}+c) = \phi(g\cdot\boldsymbol{x},g\cdot\boldsymbol{\xi}).
\end{aligned}
$$

# B. Example: Depth-2 Fully-Connected Network

Here we reproduce the ridgelet transform for depth-2 fully-connected network, originally presented in Sonoda et al. (2024a).

Set $X := \mathbb{R}^m$ (data domain), $\Xi := \mathbb{R}^m \times \mathbb{R}$ (parameter domain), and $G := \mathrm{Aff}(m) = GL(m) \ltimes \mathbb{R}^m$ be the $m$-dimensional affine group, acting on data domain $X$ by

$$
g\cdot\boldsymbol{x} := L\boldsymbol{x}+\boldsymbol{t}, \quad g=(L,\boldsymbol{t})\in GL(m)\ltimes\mathbb{R}^m, \ \boldsymbol{x}\in X.
$$

Addition to this, let $\pi$ be the regular representation of $\mathrm{Aff}(m)$ on $L^2(X)$, namely

$$
\pi(g)[f](\boldsymbol{x}) := |\det L|^{-1/2}f(L^{-1}(\boldsymbol{x}-\boldsymbol{t})), \quad f\in L^2(X) \text{ and } g=(L,\boldsymbol{t})\in GL(m)\ltimes\mathbb{R}^m.
$$

Further, define a *dual action* of $\mathrm{Aff}(m)$ on the parameter domain $\Xi$ as

$$
g\cdot(\boldsymbol{a},b) = (L^{-\top}\boldsymbol{a}, b+\boldsymbol{t}^{\top}L^{-\top}\boldsymbol{a}), \quad g=(L,\boldsymbol{t})\in GL(m)\ltimes\mathbb{R}^m, \ (\boldsymbol{a},b)\in\Xi.
$$

Then, we can see the feature map $\phi(\boldsymbol{x},(\boldsymbol{a},b)) := \sigma(\boldsymbol{a}\cdot\boldsymbol{x}-b)$ is joint-$G$-invariant. In fact,

$$
\phi(g\cdot\boldsymbol{x},g\cdot(\boldsymbol{a},b)) = \sigma\left(L^{-\top}\boldsymbol{a}\cdot(L\boldsymbol{x}+\boldsymbol{t})-(b+\boldsymbol{t}^{\top}L^{-\top}\boldsymbol{a})\right) = \sigma(\boldsymbol{a}\cdot\boldsymbol{x}-b) = \phi(\boldsymbol{x},(\boldsymbol{a},b)).
$$

Because the regular representation $\pi$ of $G = \mathrm{Aff}(m)$ is irreducible, by Theorem 3.10, the depth-2 neural network and corresponding ridgelet transform:

$$
\texttt{NN}[\gamma](\boldsymbol{x}) = \int_{\mathbb{R}^m\times\mathbb{R}}\gamma(\boldsymbol{a},b)\sigma(\boldsymbol{a}\cdot\boldsymbol{x}-b)\mathrm{d}\boldsymbol{a}\mathrm{d}b, \quad \text{and} \quad \texttt{R}[f](\boldsymbol{a},b) = \int_{\mathbb{R}^m}f(\boldsymbol{x})\overline{\rho(\boldsymbol{a}\cdot\boldsymbol{x}-b)}\mathrm{d}\boldsymbol{x},
$$

satisfy the reconstruction formula $\texttt{NN}\circ\texttt{R} = (\!(\sigma,\rho)\!)\,\mathrm{Id}_{L^2(\mathbb{R}^m)}$.

# C. $cc$-Universality by Bochner-Integral Representation

We summarize a general scheme to show the $cc$-universality by discretizing the integral representation.

## C.1. Notes on Bochner Integral

We refer to Chapter VI of Lang (1993), where the chapter title *The General Integral* refers to Banach-valued integrals.

Let $(X, \mathcal{M}, \mu)$ be a measure space, and let $E$ be a Banach space.

**Lemma C.1.** *A strongly $\mathcal{M}$-measurable function $f : X \to E$ is Bochner $\mu$-integrable iff $\|f\|$ is $\mu$-integrable, i.e.* $\int_X \|f(x)\| \mathrm{d}\mu(x) < \infty$.

See Yosida (Theorem 1, V.5 1995) for the proof.

**Lemma C.2** (Dominated Convergence Theorem for Bochner Integral)**.** *Let $\{f_n : X \to E\}$ be a sequence of Bochner integrable functions.*

- *Assume $\{f_n\}$ converges almost everywhere to some function $f : X \to E$.*

- *Let $g : X \to \mathbb{R}$ be an integrable function such that for all $n$, $\|f_n(x)\| \leq g(x)$ for almost every $x \in X$.*

*Then $f$ is Bochner integrable and $L^1$-convergent to $f$, i.e.*

$$\lim_{n \to \infty} \int_X \|f - f_n\| \mathrm{d}\mu = 0, \quad and \quad \int_X f \mathrm{d}\mu = \lim_{n \to \infty} \int_X f_n \mathrm{d}\mu$$

See Lang (Theorem 5.8, VI, § 5 1993) for the proof.

**Lemma C.3** (Bounded Convergence Theorem)**.** *Assume additionally that $X$ is a finite measure space. Let $\{f_n : X \to E\}$ be a sequence of Bochner integrable functions.*

- *Assume $\{f_n\}$ converges almost everywhere to some function $f : X \to E$.*

- *Let $G > 0$ be a uniform constant such that for all $n$, $\|f_n(x)\| \leq G$ for almost every $x \in X$.*

*Then $f$ is Bochner integrable and $L^1$-convergent to $f$, i.e.*

$$\lim_{n \to \infty} \int_X \|f - f_n\| \mathrm{d}\mu = 0, \quad and \quad \int_X f \mathrm{d}\mu = \lim_{n \to \infty} \int_X f_n \mathrm{d}\mu$$

This is an immediate corollary of DCT with dominant function being $g := G$.

## C.2. Application to Discretization of Integral Representation

**Theorem C.4** (Uniform Approximation of Integral Representation by Finite Networks)**.** *Let $X$ be a topological space, $Y$ be a Banach space, and $E := C(X; Y)$ be the Banach space of $Y$-valued continuous functions on $X$ equipped with the uniform norm $\|f\|_{C(X;Y)} := \sup_{x \in X} \|f(x)\|_Y$.*

*Let $\Xi$ be a compact metric space with finite Borel measure, let $\phi : X \times \Xi \to Y$ and $\gamma : \Xi \to \mathbb{R}$ (or $\phi : X \times \Xi \to \mathbb{R}$ and $\gamma : \Xi \to Y$) be measurable functions. Assume that the $E$-valued function $\varphi(\xi) := \gamma(\xi)\phi(\bullet, \xi)$ is Lipschitz continuous, namely*

$$L_\varphi := \sup_{\xi, \xi' \in \Xi} \frac{\|\varphi(\bullet, \xi) - \varphi(\bullet, \xi')\|_E}{d_\Xi(\xi, \xi')} < \infty.$$

*Then, the $Y$-valued continuous model*

$$\mathtt{NN}(x) := \int_\Xi \gamma(\xi)\phi(x, \xi)\mathrm{d}\xi, \quad x \in X$$

*exists in the sense of $C(X;Y)$-valued Bochner integral, and there exists a sequence of finite models: For each $n$,*

$$\mathtt{fNN}_n(x) := \sum_{i=1}^{N_n} w_{n,i}\phi(x, \xi_{n,i}), \quad x \in X$$

*with some $(w_{n,i}, \xi_{n,i}) \in \mathbb{R} \times \Xi$ (or $\in Y \times \Xi$) that uniformly converges to the continuous model, namely*

$$\lim_{n\to\infty} \|\mathtt{NN} - \mathtt{fNN}_n\|_{C(X;Y)} = 0.$$

*Remark* C.5. We do not need to assume a measure on $X$ since we only consider continuous functions with sup-norm, neither the compactness of $X$ since the Lipschitzness of $\varphi$ is simply assumed. We use the compactness of $\Xi$ to take a finite covering and obtain a finite representative points $\{\xi_{n,i}\}$, and to guarantee the existence of maximum $\sup_{\xi\in\Xi} \|\varphi(\xi)\|_E$ (with uniform continuity of $\varphi$). We use the metric of $\Xi$ to assume and use the Lipschitzness of $\varphi$ to show the a.e. convergence. We use the finiteness of $\mathrm{vol}(\Xi)$ to use the bounded convergence theorem instead of dominated convergence theorem, and to show the Bochner integrability of discretized models. We do not need to assume the Bochner integrability of $\varphi$ as it is a consequence.

*Proof.* Let $\mathcal{A}_n$ be a sequence of finite disjoint covering of $\Xi$ with diameter at most $1/n$. Namely each $\mathcal{A}_n$ is a finite family $\{A_{n,i} \mid i \in [N_n]\}$ (with cardinality $N_n < \infty$) of measurable sets satisfying $\Xi = \bigsqcup_{i=1}^{N_n} A_{n,i}$ and $\sup_{\xi,\xi'\in A_{n,i}} d_\Xi(\xi, \xi') \le 1/n$.

Put a sequence $\{\varphi_n : \Xi \to E \mid n \in \mathbb{N}\}$ of functions: For each $n$,

$$\varphi_n(\xi) := \sum_{i=1}^{N_n} \chi_{A_{n,i}}(\xi)\varphi(\xi_{n,i}), \quad \xi \in \Xi.$$

Then, (1) the sequence is uniformly bounded: For any $n$,

$$\|\varphi_n(\xi)\|_E = \left\|\sum_{i=1}^{N_n} \chi_{A_{n,i}}(\xi)\varphi(\xi_{n,i})\right\|_E \le \sum_{i=1}^{N_n} \chi_{A_{n,i}}(\xi)\big\|\varphi(\xi_{n,i})\big\|_E \le \sup_{\xi\in\Xi} \|\varphi(\xi)\|_E < \infty.$$

The last maximum exists because $\Xi$ is compact and $\varphi$ is uniformly continuous. As a consequence, (2) each function is Bochner integrable: For any $n$,

$$\int_\Xi \|\varphi_n(\xi)\|_E \mathrm{d}\xi \le \mathrm{vol}(\Xi) \sup_{\xi\in\Xi} \big\|\varphi(\xi)\big\|_E < \infty.$$

Finally, (3) it a.e.-converges to $\varphi$: At almost every $\xi \in \Xi$, there uniquely exists $A_{n,i}$ including $\xi$, so

$$\|\varphi(\xi) - \varphi_n(\xi)\|_E = \|\varphi(\xi) - \varphi(\xi_{n,i})\|_E \le L_\varphi \|\xi - \xi_{n,i}\|_E = L_\varphi/n \to 0, \quad n \to \infty.$$

Hence, by the bounded convergence theorem (Lemma C.3), $\varphi$ is Bochner integrable and

$$\int_\Xi \varphi_n(\xi)\mathrm{d}\xi = \int_\Xi \sum_{i=1}^{N_n} \chi_{A_{n,i}}(\xi)\varphi(\xi_{n,i})\mathrm{d}\xi = \sum_{i=1}^{N_n} \mathrm{vol}(A_{n,i})\gamma(\xi_{n,i})\phi(\bullet, \xi_{n,i})$$

$$\to \int_\Xi \varphi(\xi)\mathrm{d}\xi = \int_\Xi \gamma(\xi)\phi(\bullet, \xi)\mathrm{d}\xi, \quad \text{as } n \to \infty.$$

In other words, putting $w_{n,i} = \mathrm{vol}(A_{n,i})\gamma(\xi_{n,i})$ for short, we have shown

$$\lim_{n\to\infty} \left\|\sum_{i=1}^{N_n} w_{n,i}\phi(\bullet, \xi_{n,i}) - \int_\Xi \gamma(\xi)\phi(\bullet, \xi)\mathrm{d}\xi\right\|_{C(X;Y)} = 0. \quad \square$$

**Notes.** In the proof, we only used the existence of the partition with diameter at most $\varepsilon$ by the compactness. Once $\gamma$ and $\phi$ are specified, we may make use of the structures of them in a computable and effective manner. For example, a better discretization scheme is investigated in the so-called *Maurey-Jones-Barron (MJB) theory* to achieve the dimension independent *Barron's bound* (see, e.g., Kainen et al., 2013).

# D. Boundedness of M ∘ R

We present several sufficient conditions for the $L^2$-boundedness of M ∘ R.

**Lemma D.1.** *For any $f \in L^2(X;Y)$, put*

$$T[f](x) := \mathtt{M} \circ \mathtt{R}[f](x) = \int_\Xi \left[ \int_X f(y)\overline{\psi(y,\xi)}\mathrm{d}y \right] \phi(x,\xi)\mathrm{d}\xi, \quad x \in X.$$

*The $T$ is bounded, namely there exists a constant $C_T$ such that for any $f \in L^2(X;Y)$,*

$$\|T[f]\|_{L^2(X;Y)} \le C_T \|f\|_{L^2(X;Y)},$$

*when at least one of the following conditions hold:*

*T1. The integral kernel $k(x,y) := \int_\Xi \psi(y,\xi)\overline{\phi(x,\xi)}\mathrm{d}\xi$ is square integrable.*

*T2. The two feature maps $\phi, \psi$ are square integrable.*

*T3. For some topological vector space $\Gamma$, both $\mathtt{R} : L^2(X) \to \Gamma$ and $\mathtt{M} : \Gamma \to L^2(X)$ are bounded.*

*Proof.* Case T1.

$$
\begin{aligned}
\|T[f]\|^2_{L^2(X)} &= \int_X \left| \int_\Xi \left[ \int_X f(y)\overline{\psi(y,\xi)}\mathrm{d}y \right] \phi(x,\xi)\mathrm{d}\xi \right|^2 \mathrm{d}x \\
&= \int_X \left| \int_X f(y) \left[ \int_\Xi \overline{\psi(y,\xi)}\phi(x,\xi)\mathrm{d}\xi \right] \mathrm{d}y \right|^2 \mathrm{d}x \\
&= \int_X \left| \int_X f(y)\overline{k(x,y)}\mathrm{d}y \right|^2 \mathrm{d}x \\
&\le \left[ \int_X |f(y)|^2 \mathrm{d}y \right] \left[ \int_{X \times X} |k(x,y)|^2 \mathrm{d}y\mathrm{d}x \right] = \|f\|^2_{L^2(X)}\|k\|^2_{L^2(X \times X)}.
\end{aligned}
$$

Case T2.

$$
\begin{aligned}
\|T[f]\|^2_{L^2(X)} &= \int_X \left| \int_\Xi \mathtt{R}[f](\xi)\phi(x,\xi)\mathrm{d}\xi \right|^2 \mathrm{d}x \\
&\le \int_X \int_\Xi |\mathtt{R}[f](\xi)\phi(x,\xi)|^2 \mathrm{d}\xi\mathrm{d}x \\
&\le \|\phi\|^2_{L^2(X,\Xi)} \int_\Xi \left| \int_X f(y)\overline{\psi(y,\xi)}\mathrm{d}y \right|^2 \mathrm{d}\xi \\
&\le \|\phi\|^2_{L^2(X,\Xi)} \int_\Xi \int_X |f(y)\overline{\psi(y,\xi)}|^2 \mathrm{d}y\mathrm{d}\xi \\
&\le \|\phi\|^2_{L^2(X,\Xi)}\|\psi\|^2_{L^2(X,\Xi)}\|f\|^2_{L^2(X)}
\end{aligned}
$$

Case T3.

$$\|T[f]\|_{L^2(X)} = \|\mathtt{M} \circ \mathtt{R}[f]\|_{L^2(X)} \le C_\mathtt{M}\|\mathtt{R}[f]\|_\Gamma \le C_\mathtt{M}C_\mathtt{R}\|f\|_{L^2(X)}$$

$\square$

**Lemma D.2.** *Suppose $\phi_2 : X_2 \times \Xi_2 \to \mathbb{R}$ is square integrable, and $\phi_1 : X_1 \times \Xi_1 \to X_2$ has inverse jacobian $J_1(\xi_1,y) := |\det D[\phi_1](\xi_1, \phi_1^{-1}(y))|^{-1}$ satisfying $\|J_1\|_{L^1(\Xi_1) \otimes L^\infty(Y)} < \infty$. Then, $\phi_2 \circ \phi_1 : X_1 \times \Xi_{1:2} \to \mathbb{R}$ is square integrable.*

*Remark* D.3. When input and output dimensions differ, replace the jacobian $J_1$ with $\sqrt{J_1 \circ J_1^\top}$. See Evans & Gariepy (2015) for more details.

*Proof.* For almost every $\xi_1 \in \Xi_1$, put $x = \phi_1^{-1}(\xi_1, y)$, so $\mathrm{d}x = J_1(\xi_1, y)\mathrm{d}y$ with $J_1(\xi_1, y) := |\det D[\phi_1](\xi_1, \phi_1^{-1}(y))|^{-1}$. Put $Y := \bigcup_{\xi_1 \in \Xi_1} \phi_1(\xi_1, X)$. Then,

$$
\begin{aligned}
\int_{X \times \Xi_{1:2}} |\phi_2(\xi_2, \phi_1(\xi_1, x))|^2 \mathrm{d}x \mathrm{d}\xi_{1:2} &= \int_{\Xi_{1:2}} \int_{\phi_1(\xi_1, X)} |\phi_2(\xi_2, y)|^2 J_1(\xi_1, y) \mathrm{d}y \mathrm{d}\xi_{1:2} \\
&\leq \int_{\Xi_{1:2}} \left[ \int_{\phi_1(\xi_1, X)} |\phi_2(\xi_2, y)|^2 \mathrm{d}y \right] \left[ \sup_{y \in \phi_1(\xi_1, X)} J_1(\xi_1, y) \right] \mathrm{d}\xi_{1:2} \\
&\leq \int_{\Xi_{1:2}} \left[ \int_Y |\phi_2(\xi_2, y)|^2 \mathrm{d}y \right] \left[ \sup_{y \in Y} J_1(\xi_1, y) \right] \mathrm{d}\xi_{1:2} \\
&= \|\phi_2\|_{L^2(\Xi_2 \times Y)}^2 \|J_1\|_{L^1(\Xi_1) \otimes L^\infty(Y)}. \qquad \square
\end{aligned}
$$

## E. Irreducibility of the infinite-dimensional representation of the Affine group

We refer to Folland (2015, Theorem 6.42) for more details on the infinite-dimensional irreducible representations. The following proofs are attributed to Kobayashi & Oshima (2005), a Japanese textbook.

**Lemma E.1** (Kobayashi & Oshima (2005, Lemma 2.8))**.** *Let $T = \mathbb{R}^m$ be the translation group in dimension $m$, and $\tau : T \to \mathcal{U}(L^2(\mathbb{R}^m))$ be the unitary representation of $T$ defined by the regular action. A closed subspace $V$ of $L^2(\mathbb{R}^m)$ is $\tau$-invariant iff there exists a measurable set $E$ of $\mathbb{R}^m$ such that*

$$
V = \mathcal{F}^{-1}[L^2(E)].
$$

*Proof.* Let $V^\perp$ denote the orthocomplement of $V$. Since $V$ is $\tau$-invariant, for all $f \in V$ and $g \in V^\perp$,

$$
f * g^*(x) = \int_{\mathbb{R}^m} f(x - y)g^*(y)\mathrm{d}y = \int_{\mathbb{R}^m} f(x + y)\overline{g(y)}\mathrm{d}y = \langle \tau_{-x}[f], g \rangle = 0.
$$

Therefore, for all $f \in V$ and $g \in V^\perp$,

$$
\widehat{f}(\xi)\overline{\widehat{g}(\xi)} = 0, \quad \mathrm{a.\,e.}\ \xi \in \mathbb{R}^m. \tag{1}
$$

Let $\mu$ be the Gaussian measure on $\mathbb{R}^m$ defined by $\mathrm{d}\mu(\xi) := \exp(-|\xi|^2)\mathrm{d}\xi$. Put

$$
\mathcal{E}(V) := \bigcup_{f \in V} \{E \subset \mathbb{R}^m \text{ measurable} \mid E \subset \mathrm{supp}\,\widehat{f}\},
$$

and

$$
A := \sup_{E \in \mathcal{E}(V)} \mu(E) \leq \mu(\mathbb{R}^m) < \infty.
$$

Let $E_j$ be a sequence of measurable sets of $\mathcal{E}(V)$ satisfying $\lim_{j \to \infty} \mu(E_j) = A$, and set $F := \bigcup_j E_j$. Then $F$ is measurable, and $\mu(F) \geq A$ because $\limsup \mu(E_j) \leq \mu(\limsup E_j)$ in general. By Equation (1), $\widehat{g}(\xi) = 0$ at a.\,e. $\xi \in E_j$ for all $j$ and $g \in V^\perp$. Hence

$$
V^\perp \subset \mathcal{F}^{-1}\left[L^2(\mathbb{R}^m \setminus F)\right],
$$

thus

$$
V \supset \left(\mathcal{F}^{-1}\left[L^2(\mathbb{R}^m \setminus F)\right]\right)^\perp \supset \mathcal{F}^{-1}\left[L^2(F)\right].
$$

Conversely, suppose that there exist an $f \in V$ and a measurable subset $F' \subset \mathbb{R}^m \setminus F$ satisfying $\widehat{f}(\xi) \neq 0$ a.\,e. $\xi \in F'$. Then $\widehat{g}(\xi) = 0$ at a.\,e. $\xi \in F'$ for all $g \in V^\perp$, and we have

$$
V^\perp \subset \mathcal{F}^{-1}\left[L^2(\mathbb{R}^m \setminus (F \cup F'))\right],
$$

thus

$$V \supset \left( \mathcal{F}^{-1} \left[ L^2(\mathbb{R}^m \setminus (F \cup F')) \right] \right)^{\perp} \supset \mathcal{F}^{-1} \left[ L^2(F \cup F') \right],$$

which means $F \cup F' \in \mathcal{E}(V)$. By the definition of $A$, $A \geq \mu(F \cup F') = \mu(F) + \mu(F')$. On the other hand, $\mu(F) \geq A$. So, $\mu(F') = 0$, but this implies $\widehat{f}(\xi) = 0$ a. e. $\xi \in \mathbb{R}^m \setminus F$. Therefore

$$V \subset \mathcal{F}^{-1} \left[ L^2(F) \right],$$

thus

$$V = \mathcal{F}^{-1} \left[ L^2(F) \right].$$

$\square$

**Theorem E.2** (Kobayashi & Oshima (2005, Theorem 2.10)). *Let $G = \mathrm{Aff}(\mathbb{R}^m) \cong GL(m, \mathbb{R}) \ltimes \mathbb{R}^m$ be the affine group. For any $\lambda \in \mathbb{C}$, let $\pi$ be the representation of $G$ on $L^2(\mathbb{R}^m)$ defined by*

$$\pi_g[f](x) := |\det A|^{-\lambda} f(g^{-1} \cdot x), \quad x \in \mathbb{R}^m, g = (A, b) \in GL(m, \mathbb{R}) \ltimes \mathbb{R}^m.$$

*Additionaly, let $\widehat{\pi}$ be another representation of $G$ on $L^2(\mathbb{R}^m)$ defined by*

$$\widehat{\pi}_g[h](\xi) := e^{ib \cdot \xi} |\det A|^{1-\lambda} h(A^{\top} \xi), \quad \xi \in \mathbb{R}^m, g = (A, b) \in GL(m, \mathbb{R}) \ltimes \mathbb{R}^m$$

*so that $\mathcal{F}$ intertwines $\pi$ and $\widehat{\pi}$, namely $\mathcal{F} \circ \pi_g = \widehat{\pi}_g \circ \mathcal{F}$ for all $g \in G$. Suppose $\lambda \in \frac{1}{2} + i\mathbb{R}$, then $\pi$ is an irreducible unitary representation of $G$ on $L^2(\mathbb{R}^m)$.*

*Proof.* Note that $\pi$ is unitary iff $\lambda \in \frac{1}{2} + i\mathbb{R}$. We identify the translation group $T = \mathbb{R}^m$ as a subgroup of the affine group $G = \mathrm{Aff}(\mathbb{R}^m)$ by embedding $b \in T$ to $(I, b) \in G$. Suppose $V \neq 0$ be a $\pi$-invariant closed subspace of $L^2(\mathbb{R}^m)$. Then it is also $\tau$-invariant because $\pi(I, b) = \tau(b)$. Hence by Lemma E.1, there exists a measurable subset $E$ of $\mathbb{R}^m$ satisfying $V = \mathcal{F}^{-1} \left[ L^2(E) \right]$. Further, the image $\mathcal{F}(V) = L^2(\mathbb{R}^m)$ is $\widehat{\pi}$-invariant. Therefore by the definition of $\widehat{\pi}$, $E$ has to be invariant (except a measure zero subset) under the dual action of $A \in GL(m, \mathbb{R})$ given by

$$\xi \mapsto A^{-\top} \xi, \quad \xi \in \mathbb{R}^m.$$

But this action is known to be ergodic, that is, such a measurable set is either $\mathbb{R}^m$ or . Hence $E = \mathbb{R}^m$ and $V = \mathcal{F}^{-1} \left[ L^2(E) \right] = L^2(\mathbb{R}^m)$, which yields the irreducibility of $\pi$. $\square$

