# OpenReview forum: "Deep Ridgelet Transform and Unified Universality Theorem for Deep and Shallow Joint-Group-Equivariant Machines"
_ICML.cc/2025/Conference — ICML 2025 poster_

### Official Review · Reviewer_Jwhk · 2025-03-09

**Overall Recommendation:** 4

**Summary:**

The paper studies so called "joint-equivariant networks", which are neural networks that are simultaneously equivariant with respect to a group action on the input data as well as the parameter space. This can be viewed as a generalization of previously studied group-equivariant neural network, in the sense that the joint-equivariant networks reduces to the aforementioned networks upon taking a trivial group action on the parameter space. This framework also includes standard fully connected neural networks, which are not equivariant. The main result is a universality theorem for the joint-equivariant that unifies certain previous universality results in the literature.

## update after rebuttal
I stand by my original score.

**Claims And Evidence:**

Yes, see below.

**Essential References Not Discussed:**

It strikes me as strange that the authors do not cite the original references for group equivariant CNNs by Cohen et al. They mention one paper from 2019 by Cohen, Geiger and Weiler which provides some of the underlying mathematical structure of GCNNs. However, there are several papers before that introducing the key structure of GCNNS, such as

Group Equivariant Convolutional Networks - Cohen and Welling
https://arxiv.org/abs/1602.07576


Steerable CNNs - Cohen and Welling
https://arxiv.org/abs/1612.08498


Spherical CNNs - Cohen, Geiger, Koehler and Welling
https://arxiv.org/abs/1801.10130

**Experimental Designs Or Analyses:**

NA

**Methods And Evaluation Criteria:**

NA

**Other Comments Or Suggestions:**

To reiterate the comment above: How are your depth-n G-convolutional layers related to the group convolutional layers by Cohen et al?

**Other Strengths And Weaknesses:**

In section 6 they discuss G-convolutional networks of Depth n. However, nowhere do they cite the original papers on group convolutional layers mentioned above. They also do not discuss how their result relate to these previous results. I would certainly have liked to see a detailed discussion of how their GCN-layer is related to previously constructed group equivariant layers by Cohen et al. I think this is a weakness of the paper.

**Questions For Authors:**

See above

**Relation To Broader Scientific Literature:**

See below

**Theoretical Claims:**

Yes, it is my opinion that the claims in the paper are supported by convincing evidence. The theorems are proven rigorously and the mathematical framework is consistent.

It is worth mentioning that the same group of authors have written several papers on this topic before. Although the papers address related points the present paper represents sufficiently novel results to warrant publication on its own.

---

> ### Author Rebuttal · Authors · 2025-03-27
>
> > How are your depth-n G-convolutional layers related to the group convolutional layers by Cohen et al?
>
> Thank you for your question regarding concrete examples. We fully acknowledge the pioneering contributions of Cohen and colleagues in the development of group-equivariant neural networks.
>
> The focus of this study lies in universality and a general geometric perspective. Therefore, in this paper, we chose to highlight a single representative example with a particular emphasis on differential geometric aspects, specifically the 2019 paper.
>
> The relationship between the ridgelet transform of group-convolution networks and existing works—especially in the context of universal approximation theorems—is thoroughly discussed in Sections 5 and 6 of Sonoda et al. (2022a). We kindly refer the reviewer and future readers to these sections for further details.
>
> The three papers suggested by the reviewer do not explicitly analyze the universality or expressive power of the proposed networks. The group convolutions in Group Equivariant Convolutional Networks and Steerable CNNs correspond to group convolutions where a finite group $G$ acts on $X = \mathbb{R}^{n \times n \times k}$ for some $n$ and $k$, and thus their universality follows as a corollary of Theorem 6.1. Similarly, the convolution in Spherical CNNs corresponds to the case where compact group $G = SO(3)$ acts on $X = \mathrm{SO}(3)$ (or homogeneous space $S^2$), and thus the universality can also be shown in the sense of Theorem 6.1.

---

> > ### Comment · Reviewer_Jwhk · 2025-04-04
> >
> > Thanks for the clarifications!

---

### Official Review · Reviewer_9cKy · 2025-03-12

**Overall Recommendation:** 3

**Summary:**

The main result of this paper is Theorem 3.10, which provides a closed form formula for a ridgelet transform of learning machines with joint-group-equivariant maps. Previous works have been deriving the closed form formula for the ridgelet transforms of depth-2 networks. This paper has generalized these results to the wider class of neural networks that covers joint-equivariant machines, fully connected networks, group-convolutional networks, and a depth-2 network with quadratic forms.

**Claims And Evidence:**

The claims made in the submission are supported by clear and convincing evidence. All theorems and lemmas are provided with complete proofs.

**Essential References Not Discussed:**

Essential related works on ridgelet transforms and universal approximation theorems are included in the introduction section.

**Experimental Designs Or Analyses:**

No experiment is provided.

**Methods And Evaluation Criteria:**

No method nor evaluation criteria is proposed in this paper.

**Other Comments Or Suggestions:**

See weaknesses.

**Other Strengths And Weaknesses:**

**Strengths**

- The paper successfully provides a closed form formula for a ridgelet transform of learning machines with joint-group-equivariant maps, thus proposes a unified theoretical framework for understanding universal approximation in both deep and shallow networks, addressing a fundamental topic in machine learning.

- The results are strongly supported by rigorously proofs, making a valuable theoretical contribution.

- Several concrete examples are provided to demonstrate the applicability of the main theorem, enhancing its relevance to various machine learning architectures.

**Weaknesses**

- The paper relies on dense mathematical formalism, making it less accessible to a broader audience, especially those outside mathematical machine learning.

- The paper has no numerical experiments or empirical validation as it focuses entirely on theoretical derivations. Personally, I think this paper is more suitable for mathematics community than machine learning community.

- The authors mention several times that this result unifies the universal approximation theorem. But it is not clear how Theorem 3.10 can imply the universal approximation theorem, say the one established by (Pinkus, 1999).

Pinkus, Allan (January 1999). "Approximation theory of the MLP model in neural networks". Acta Numerica. 8: 143–195.

**Questions For Authors:**

How does Theorem 3.10 imply the traditional universality approximation theorem in (Pinkus, 1999)?

**Relation To Broader Scientific Literature:**

This paper provides a closed form formula for a ridgelet transform of learning machines with joint-group-equivariant maps. This result is already a great generalization of previous results in the literature on ridgelet transforms of neural networks.

**Theoretical Claims:**

I checked some of the main proofs but did not go through all of them in details. The proofs seem to be correct.

---

> ### Author Rebuttal · Authors · 2025-03-27
>
> > The authors mention several times that this result unifies the universal approximation theorem. But it is not clear how Theorem 3.10 can imply the universal approximation theorem, say the one established by (Pinkus, 1999).
>
> There is no unique definition of "universality." In the context of machine learning, the term "universality" typically corresponds to what is referred to as "density" in standard mathematical terminology. Accordingly, there exist various types of universality depending on the choice of topology.
>
> For example, the notion treated in Pinkus (1999) corresponds to $cc$-universality, or density in the topology of compact convergence, which is also known as the topology of uniform convergence on compact sets. In contrast, the type of universality we have demonstrated is $L^2$-universality, or density in the $L^2$-topology.
>
> The relationships among various notions of universality commonly used in machine learning theory are discussed in detail in the following reference:
>
> Sriperumbudur, Fukumizu, and Lanckriet. Universality, Characteristic Kernels and RKHS Embedding of Measures, JMLR, 2011.
>
> Moreover, it is generally known that once an $L^2$-universal integral representation is established, $cc$-universality of a finite-width model can be shown by employing quasi-Monte Carlo integration (more precisely, by applying the law of large numbers). Therefore, the claims of our work indeed imply $cc$-universality as well.

---

### Official Review · Reviewer_ETza · 2025-03-13

**Overall Recommendation:** 3

**Summary:**

This paper introduces a framework for proving constructive universal approximation theorems for general classes of neural networks. By invoking ideas from representation theory, the paper generalizes the ridgelet transform to a larger class of models which satisfy "joint G equivariance." As a consequence, the paper establishes universal approximation theorems for deep fully connected networks, deep group convolutional networks, and a depth 2 network on a quadratic form.

**Claims And Evidence:**

The claims in this paper (i.e the main result Theorem 3.10, and the following examples in sections 4-7), are indeed supported by clear evidence.

**Essential References Not Discussed:**

N/A

**Experimental Designs Or Analyses:**

N/A

**Methods And Evaluation Criteria:**

This paper is theoretical, and so the "methods" (i.e theorems proven) do indeed make sense for the problem at hand.

**Other Comments Or Suggestions:**

- From a clarity perspective, the machinery developed in Section 3 is quite technical and at times difficult to follow (especially for readers less familiar with representation theory). The paper could be improved by motivating the various definitions in Section 3 with a concrete example, such as the deep feedforward network.

**Other Strengths And Weaknesses:**

Strengths:
- The framework developed by this paper seems to be very general purpose, and can allow for constructive proofs of universal approximation theorems for other architectures in the future.
- To the best of my knowledge, the contribution of this paper appears to be novel.

Weaknesses:
- This paper only handles the infinite width (i.e continuous) setting, and does not provide any quantitative results on how much width is needed for a certain architecture to approximate a target function, akin to the Barron norm in (Barron, 1993).
- The implications of this work (such as depth separations between different classes of architectures) is not clear, which limits the significance.

**Questions For Authors:**

- Can the authors please address my comments in the weaknesses section above?
- Additionally, the universal approximation result for deep feedforward networks in Section 5 is unsurprising, given the universal approximation guarantee for two-layer networks and the fact that deep networks are a strictly larger hypothesis class. Could the authors please comment further on this point?

**Relation To Broader Scientific Literature:**

This paper unifies prior works on universal approximation of neural networks via the integral representation approach (i.e Barron, 1993), and generalizes prior works on closed form solutions for the ridgelet transform (Sonoda et al., 2021, 2022a, 2022b, 2024a, 2024b).

**Theoretical Claims:**

The theoretical claims appear to be sound.

---

> ### Author Rebuttal · Authors · 2025-03-27
>
> > This paper only handles the infinite width (i.e continuous) setting, and does not provide any quantitative results on how much width is needed for a certain architecture to approximate a target function, akin to the Barron norm in (Barron, 1993).
>
> In our main result, the inverse operator for the integral representation is explicitly obtained. This allows us to apply the so-called Barron’s argument to evaluate the approximation error using the Maurey–Jones–Barron (MJB) bound. In general, applying the MJB bound yields an approximation error of $O(1/\sqrt{n})$ when using a finite-width model with $n$ units.
>
> > The implications of this work (such as depth separations between different classes of architectures) is not clear, which limits the significance.
>
> (Please also refer to our fourth response for related discussion.)
>
> In addition, algebraic methods such as Schur's lemma offer a significant advantage: they allow us to assess density (or universality) even when the input and output are not vectors. While there are several classical theorems—such as the Stone–Weierstrass and Hahn–Banach theorems—that provide general tools for determining universality, applying them to neural networks with deep hierarchical structures requires substantial effort and ingenuity.
>
> For example, proof techniques that trace back to Hornik et al. (1989) often rely on repeatedly differentiating activation functions to construct polynomial approximations. While these techniques are mathematically valid, they tend to obscure the underlying mechanism by which neural networks perform approximation, and may not offer much intuitive understanding.
>
> In contrast, conditions such as *joint group equivariance* and tools like the *ridgelet transform*, which serves as an inverse operator, provide a more direct and interpretable explanation of how neural networks approximate functions.
>
> > The paper could be improved by motivating the various definitions in Section 3 with a concrete example, such as the deep feedforward network.
>
> Thank you for the suggestion. We will incorporate a concrete example, such as the deep fully connected network, immediately after the definition to improve clarity.
>
> > Additionally, the universal approximation result for deep feedforward networks in Section 5 is unsurprising, given the universal approximation guarantee for two-layer networks and the fact that deep networks are a strictly larger hypothesis class. Could the authors please comment further on this point?
>
> Thank you for your request. The value of this work does not lie solely in the illustrative example presented in Section 5. Rather, it lies in providing a unified, simple, and constructive principle for analyzing the expressive power of a wide range of learning machines, as demonstrated in Sections 4 through 7.
>
> In deep learning, diverse architectures are proposed on a daily basis. Traditionally, analyzing the expressive power of each new architecture required manual, case-by-case efforts by experts. This study identifies a key condition—*joint-group-equivariance*—that enables such analysis to be performed systematically. We believe this is where the main contribution of our work lies.

---

### Official Review · Reviewer_RWHG · 2025-03-14

**Overall Recommendation:** 4

**Summary:**

The goal of the paper is to subscribe a general framework for proving universality type theorems for a generalized class of models, that the authors call joint-group-equivariant machines. Joint-group-equivariant machines are models consisting of a sequence of joint-group-equivariant features maps, maps from the joint input-parameter space to an output (feature) space that are group equivariant (jointly in the input and parameter space). Central to the proof is an interesting application of Schur’s lemma on irreducible representations, which concludes that a certain construction for a Ridgelet transform (which is introduced in the paper) is indeed one the “dual” which maps functions of a certain class to parameters of the joint-group-equivariant model.

The authors proceed to exemplify the proof for various particular types of models, such as “Depth-n Fully-Connected
Networks”, “ Depth-n Group Convolutional Networks” and a “Quadratic-form with Nonlinearity”.

**Claims And Evidence:**

This paper is about a pure theoretical construction and universal approximation results on them, empirical evidence is not applicable.

**Essential References Not Discussed:**

See above

**Experimental Designs Or Analyses:**

N/A

**Methods And Evaluation Criteria:**

N/A

**Other Comments Or Suggestions:**

Small typo (double ‘the’) on L30, right column

**Other Strengths And Weaknesses:**

The unification under the generalized model is theoretically appealing, however I am not sure if the generalized model (joint-group-equivariant machines) are any further useful. It is not clear to me what is the impact of the unification of mostly known results. Have we learnt something insightful about machine learning, or generalization, in the process of unifying universality theorems?

**Questions For Authors:**

The depth-N fully connected neural network described in Section 5 is different that what the community is used to and it looks more general. Does the provided formulation of fully-conntected NNs generalize a ReLU layer for example?

**Relation To Broader Scientific Literature:**

I am not up-to-date with the literature on this topic

**Theoretical Claims:**

The theoretical claims seem reasonable, the writing is easy to follow and introduces clearly the constructions about which the proofs is about. On times the paper uses abstract mathematical language to give alternative descriptions to constructions which may obfuscate the reading (for example some category-theoretical parallels, whose validity I cannot testify). However, I find them redundant/complementary to the actual message of the paper, and conditioned that they are true, they do not alter its story.

---

> ### Author Rebuttal · Authors · 2025-03-27
>
> > On times the paper uses abstract mathematical language to give alternative descriptions to constructions which may obfuscate the reading (for example some category-theoretical parallels, whose validity I cannot testify). However, I find them redundant/complementary to the actual message of the paper, and conditioned that they are true, they do not alter its story.
>
> Thank you for the suggestion. We will retain the mathematically detailed supplementary explanation as an optional message intended for readers with a strong background in algebra or representation theory. Since it is strictly supplementary, skipping it will not affect the main narrative or the overall understanding of the paper.
>
> > The unification under the generalized model is theoretically appealing, however I am not sure if the generalized model (joint-group-equivariant machines) are any further useful. It is not clear to me what is the impact of the unification of mostly known results. Have we learnt something insightful about machine learning, or generalization, in the process of unifying universality theorems?
>
> The main theorem of this work is not merely a unification of existing results; a key strength lies in its ability to systematically and mechanically assess universality even for novel architectures. For example, it enables the straightforward design of expressive networks with universal approximation capability, such as *quadratic networks*.
>
> In the era of large language models, the diversity of proposed architectures has significantly increased. Manually analyzing the expressive power of each new model on a case-by-case basis is no longer feasible from a theoretical standpoint. Our framework addresses this challenge by offering a general principle.
>
> In particular, algebraic tools such as *Schur's lemma* provide a notable advantage in that they allow us to assess universality (or density) even when the inputs and outputs are not vectors—something that traditional techniques often struggle with. While classical theorems like the Stone–Weierstrass or Hahn–Banach theorems offer generic tools for establishing universality, applying them to deep neural networks with hierarchical structures requires considerable ingenuity.
>
> By contrast, proof techniques originating with Hornik et al (1989), which rely on repeatedly differentiating activation functions to construct polynomial approximations, often obscure the intuitive mechanism by which neural networks perform approximation. In comparison, our use of *joint group equivariance* and *ridgelet transforms* as inverse operators provides a more direct and interpretable understanding of the approximation process in deep networks.
>
> Regarding generalization: Theoretical analysis of generalization error is often based on estimates of the Rademacher complexity of the hypothesis class. However, in deep learning, the Rademacher complexity typically scales exponentially with network depth. This is at odds with practical observations, where deeper networks tend to generalize better.
>
> This discrepancy arises from the limitations of current techniques for analyzing composite functions: the reliance on coarse Lipschitz estimates leads to overly pessimistic bounds. To address this gap, we revisited expressive power analysis and developed the *deep ridgelet transform* as a theoretical framework that can precisely handle compositions of functions.
>
> > The depth-N fully connected neural network described in Section 5 is different that what the community is used to and it looks more general. Does the provided formulation of fully-conntected NNs generalize a ReLU layer for example?
>
> Yes, the presented example extends typical networks, and especially it includes the case when $\sigma$ is ReLU.

---

### Decision · Program_Chairs · 2025-05-01

**Decision:**

Accept (poster)

**Comment:**

The reviews are all positive. The method proposes a novel and more general architecture to leverage symmetries, as supported by a complete and rigorous theoretical analysis. I am in favor of accepting this paper.